# Power sector investment implications of climate impacts on renewable resources in Latin America and the Caribbean

Silvia R. Santos da Silva [1,2✉], Mohamad I. Hejazi[2], Gokul Iyer [2], Thomas B. Wild [2,3], Matthew Binsted [2], Fernando Miralles-Wilhelm [1,2,3], Pralit Patel [2], Abigail C. Snyder [2] & Chris R. Vernon [4]

Climate change mitigation will require substantial investments in renewables. In addition, climate change will affect future renewable supply and hence, power sector investment requirements. We study the implications of climate impacts on renewables for power sector investments under deep decarbonization using a global integrated assessment model. We focus on Latin American and Caribbean, an under-studied region but of great interest due to its strong role in international climate mitigation and vulnerability to climate change. We find that accounting for climate impacts on renewables results in significant additional investments ($12–114 billion by 2100 across Latin American countries) for a region with weak financial infrastructure. We also demonstrate that accounting for climate impacts only on hydropower—a primary focus of previous studies—significantly underestimates cumulative investments, particularly in scenarios with high intermittent renewable deployment. Our study underscores the importance of comprehensive analyses of climate impacts on renewables for improved energy planning.

[1] Department of Atmospheric and Oceanic Science, University of Maryland, College Park, MD, USA. [2] Joint Global Change Research Institute, Pacific Northwest National Laboratory, College Park, MD, USA. [3] Earth System Science Interdisciplinary Center, College Park, MD, USA. [4] Pacific Northwest National Laboratory, Richland, WA, USA. ✉email: silviameteoro@gmail.com

After the 2015 Paris Agreement, nations worldwide have pursued climate change mitigation strategies in the form of nationally determined contributions (NDCs) and long-term strategies (LTSs). These strategies typically include substantial renewable energy (RE) deployment[1–3]. Nevertheless, climate change might influence RE generation through long-term alterations in various environmental conditions. For example, climate change could affect biomass crop yields and hence biomass potential[4]. Likewise, climate change could affect streamflow, with implications for hydroelectricity generation[5]. Solar power production may be impaired by reduced surface solar radiation[6], or could increase (e.g., concentrating solar power) or decrease (e.g., photovoltaics) with rising air temperatures[7–9]. Wind power production could be affected by changing wind speed and air density patterns[10,11]. Hence, planners need to account for climatic impacts on RE during capacity development planning to ensure power system reliability, which is particularly relevant in the context of decarbonization strategies centered on RE expansion.

Most decarbonization scenarios (e.g., those reviewed by the Intergovernmental Panel on Climate Change (IPCC)[12]) suggest that large investments in renewables will be required, particularly under assumptions of limited or no deployment of carbon capture and storage (CCS) and nuclear technologies[13]. In this context, there is an open question about how climate impacts on renewable resources—such as those described above—could alter the understanding of the economic implications and investment needs suggested by alternative decarbonization pathways. Research on this question has been very limited and the majority of mitigation scenarios in the literature do not account for the impacts of climate change. This is the case of the about 900 mitigation scenarios reviewed in the IPCC's Fifth Assessment Report (AR5)[12]. Even the few studies exploring climate impacts within the context of decarbonization scenarios have focused only on hydropower without a comprehensive analysis of impacts on all renewable sources.

With growing literature highlighting that the energy sector, including RE production, may face serious impacts due to climate change[14–16], there have been efforts to incorporate climate impacts on renewables into energy and integrated assessment models (IAMs) to support decision-making. Methodologically, many of these studies rely on detailed process-based models (for example, hydrologic models, crop models, general circulation models (GCMs)) capable of simulating climate-impacted environmental responses that are used to modify IAM parameters linked to RE production. However, hydropower—the renewable that currently contributes the most to the global electricity supply[4]—has received considerably larger attention from the IAM literature and climate-impact studies in general (see the literature referenced in Yalew et al.[14], Solaun and Cerdá[15], Cronin et al.[16], and Emodi et al.[17]). IAM-based studies on climate impacts on hydropower (some of them conducted in the context of decarbonization scenarios as mentioned earlier) have been useful in exploring climate change implications for electricity production and capital investments[18–24]. Another group of IAM-based studies has addressed impacts on the agriculture sector (which affect biomass potential) by incorporating biophysical crop yield changes[25–28]. Regarding the representation of climate impacts on solar and wind resources in IAMs, research efforts are still incipient, and to the best of our knowledge, limited to only two studies[24,29]. Consequently, there is a gap in the literature on a comprehensive analysis of climate impacts on all renewable resources and their implications for electricity sector investments. Studies that focus on climate impacts on individual resources do not account for the compounding effects of climate impacts on multiple renewable sources and may thus under- or overestimate

investment requirements. Another gap in the literature is the lack of regionally-focused studies[15,16]. While global studies are useful in characterizing the scale of a problem, policy decisions are made at national to sub-national scales. Hence, regional analyses with a focus on national issues and circumstances are important to enhance the relevance of the analyses to decision-makers. Our study fills both of the above gaps.

We incorporate climate impacts on all renewables, namely, hydropower, biomass, solar, and wind, within the Global Change Analysis Model (GCAM)[30], a state-of-the-art global IAM. Using this improved version of the model, we examine changes in electricity generation patterns and future investment needs under decarbonization scenarios. For the purposes of this study, we focus on Latin American and the Caribbean (LAC), a greatly under-studied region despite its global relevance. For instance, in 2017, RE represented about 56% of LAC's electricity generation versus a global average of 26% (ref. [31] and Supplementary Fig. 1). Hydropower and bioenergy have dominated the regional RE portfolio, however, solar and wind have experienced rapid growth in installed capacity from 0.79 to 27.31 GW between 2008 and 2017[32]. This growth is expected to continue due to strong policies[33], and the strategic role of RE in many LAC countries' climate goals. As summarized in Supplementary Table 1, 25 LAC countries intend to foster RE as a route for mitigation. Among major economies, Brazil plans to promote non-hydropower sources, Mexico intends to focus on wind, solar photovoltaics (including distributed systems) and hydropower, and Argentina is particularly interested in promoting biofuels.

Despite the increasingly important role of RE in LAC, notably in the electricity sector, current regional consumption of fossil fuels remains a challenge for climate mitigation (fossil fuels represented roughly 70% of total primary energy supply (TPES) in 2015[34]). TPES in LAC depends primarily on oil, natural gas, bioenergy, and hydropower[34] with large oil and biofuels use in the transportation sector, while hydropower, natural gas, and oil comprise most of the electricity supply (Supplementary Fig. 2). However, at sub-regional/country levels, important departures from the overall LAC profile exist. For example, regarding hydropower, which dominates regional electricity mixes (notably in Uruguay, Brazil, and Colombia), except for Mexico, Central America, and the Caribbean and Argentina where main generating sources are natural gas and oil (Supplementary Fig. 3). Absent efforts to constrain emissions, fossil technologies in LAC are projected to expand[35] (Supplementary Figs. 2—3 provide projections from the GCAM Baseline (No Policy) scenario, which assumes no emissions mitigation actions throughout the 21st century). In light of this, previous LAC decarbonization scenarios agree that renewables, mainly biomass, solar, and wind as well as CCS technologies applied to fossil fuels and biomass are critical to mitigate energy-sector emissions, with nuclear energy typically playing less relevant roles[36–41]. In these scenarios, hydropower remains important, but its contribution to regional total generation falls over time, as hydropower capacity expansion is not expected to follow growing demands[15].

Under future climate change conditions, RE production in LAC will potentially face several challenges. By the end of the 21st century, multi-model projections using the representative concentration pathways (RCPs)[12] show mean warming levels reaching 0.6 °C to 2.0 °C in RCP2.6 and 2.2 °C to 7.0 °C in RCP8.5, and both positive and negative rainfall anomalies across the region[42]. Although there is large uncertainty intrinsic to these climate projections, their effect on future estimates of hydropower potential is manifested in terms of a strong regional variability of impacts from gains in Uruguay and the southernmost basins of Brazil to losses in northern Brazil, Colombia, northern South America, Argentina, and southern South America[20,43–45]. The

**Table 1 Scenarios explored in this study.**

| | Technology availability | | |
| --- | --- | --- | --- |
| | **FullTech** | **NoCCS & NoNewNuc** | **Baseline** |
| **Climate Impacts** | | | |
| None | RCP26_FullTech: No-climate impacts | RCP26_NoCCS & NoNewNuc: No-climate impacts | RCP60_Baseline: No-climate impacts |
| Hydropower | RCP26_FullTech: Hydropower | RCP26_NoCCS & NoNewNuc: Hydropower | RCP60_Baseline: Hydropower |
| All renewables | RCP26_FullTech: Combined impacts | RCP26_NoCCS & NoNewNuc: Combined impacts | RCP60_Baseline: Combined impacts |
| **Climate Mitigation** | RCP2.6 | RCP2.6 | RCP6.0 |

limited literature focusing on LAC suggests increased wind and solar resource potentials in Brazil[43,46–48], and, possibly, a positive general response of the main LAC bioenergy feedstock, sugarcane, to regional climatic changes[42]. For more details see Supplementary Note 1 (which briefly reviews main climate-attributable effects on RE alongside anticipated impacts for LAC). Despite its large socioeconomic and physical vulnerability to climate change, LAC has been poorly covered by energy-sector impact studies which are either global in scope or largely focused on Europe and North America[15–17].

To account for compounding climate impacts on renewables, we explore 9 illustrative scenarios using GCAM (Table 1). The scenarios vary across three dimensions, namely, assumptions about the level of climate change mitigation, climate impacts on renewables and technology availability. Along the first dimension, two scenario variants exist. We explore scenarios with no explicit climate policy, which lead to a radiative forcing of 6.0 W/m$^2$ at the end of the century. These scenarios are based on the GCAM Baseline (No Policy) scenario mentioned earlier (note that the RCP60_Baseline: No-climate impacts scenario shown in Table 1 is identical to the GCAM Baseline (No Policy) scenario). We also explore scenarios with greenhouse gas mitigation policies to reduce radiative forcing. These scenarios assume that countries across the globe (including those in LAC) achieve their NDC commitments through 2030. Beyond 2030, the scenarios assume globally coordinated mitigation efforts compatible with limiting end-of-century temperature rise to 2 °C and with the RCP2.6 (Supplementary Note 4).

Along the climate impacts dimension, we explore three variations. The first variation, named No-climate impacts, assumes no climate impacts on renewable resources, serving as a reference against which to compare the other scenarios. The Hydropower scenarios assume climate impacts on hydropower only, allowing a comparison with the approach of prior studies that have investigated electricity-sector implications due to climate impacts on hydropower[18–24]. The Combined impacts scenarios assume climate impacts on all renewable resources. Specifically, we include climate impacts on agricultural productivity (though changes in crop yields), hydropower production, and wind and solar supply-curves (Methods and Supplementary Note 3) in combination. The climate inputs for our simulations are based on bias-corrected projections from the GFDL-ESM2M, HadGEM2-ES, and IPSL-CM5A-LR general circulation models (GCMs) obtained from the Inter-Sectoral Impact Model Intercomparison Project (ISIMIP)[49,50] under the RCPs 2.6 and 6.0. Although we conduct three distinct model simulations for each climate impacts scenario corresponding to climate inputs from each of the GCMs above, we focus on mean values across all GCMs in the rest of the paper. Results based on climate model uncertainty are presented in the Supplementary Figs. 47—48. Note that the RCP2.6 is the lowest projected warming level among the RCPs considered within the IPCC AR5 and ISIMIP, and is consistent with a global warming likely below 2 °C above pre-industrial temperatures[12]. The RCP2.6 allows climate impacts on renewables being studied

in a context of strong climate change mitigation with substantial upscaling of renewable energy. On the other hand, the RCP6.0 represents a high emissions pathway[12].

Along the technology availability dimension, we explore three variations. The Baseline and FullTech scenarios assume that the full suite of power sector technologies represented by GCAM is available globally. However, the FullTech scenario includes CCS technologies that are only deployed in the context of decarbonization. The NoCCS & NoNewNuc scenario assumes no deployment of CCS technologies globally, and no new deployment of nuclear technologies in LAC. The NoCCS & NoNewNuc scenario represents a high renewable scenario—which is important within the context of LAC where future mitigation strategies are expected to rely heavily on renewables. These scenarios are consistent with many prior mitigation studies[36,37,51].

## Results

**Implications for electricity generation patterns.** Consistent with prior literature on LAC decarbonization scenarios[36–41], our mitigation RCP26 scenarios entail a significantly larger use of low-carbon energy sources and increased electrification of end-use sectors compared with a Baseline energy technology pathway (Supplementary Figs. 3-10). The RCP26_FullTech family of scenarios represents a diverse array of low-carbon technologies with bioenergy and natural gas plants equipped with CCS playing central roles in mitigation by supplanting the role of fossil-fuel-based power generation, particularly, of natural gas, through 2100 (Supplementary Figs. 5, 7 and 9). Under the RCP26_NoCCS & NoNewNuc scenarios, emissions reductions in the power sector are achieved largely through the addition of solar and wind plants (Supplementary Figs. 6, 8, and 10). Uruguay stands out for a Baseline profile already predominantly reliant on RE, particularly on wind (Supplementary Fig. 3). In Uruguay, the mitigation scenarios lead to a replacement of bioenergy without CCS by bioenergy with CCS or wind depending on the technology pathway (Supplementary Figs. 7—10). As noted below, each energy technology pathway offers distinct technological alternatives for adaptation to climate impacts on RE.

Figure 1 provides an overview of the mean differences in electricity generation for the six climate-impact scenarios relative to the reference No-Climate impacts cases. A comparison between the Combined impacts and the Hydropower scenarios highlights the possibility of an incomplete understanding of the implications of climate change on the power sector without an integrated framework that accounts for impacts on all renewables. Such an issue is apparent in most subregions for two reasons. First, some LAC subregions (particularly Brazil, S. Am. N., and S. Am. S) show nontrivial responses induced by the climate-impacted wind supply-curves (that is, considerable changes in wind power production relative to the climate-impacted hydro-electricity generation; see Supplementary Figs. 11—14 and Supplementary Note 6 for details on how multiple interacting climate impacts combine and affect the modeled RE production).

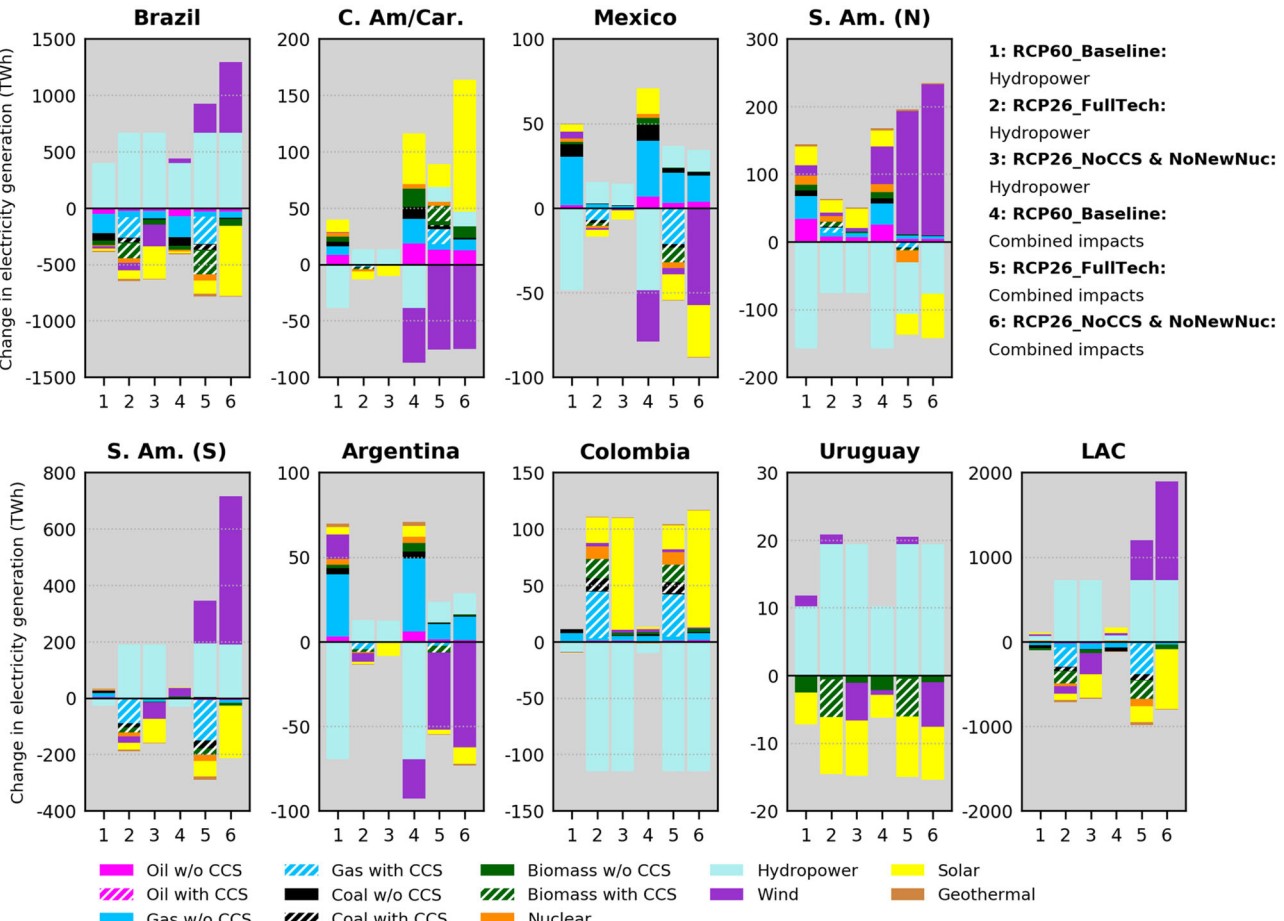

**Fig. 1 Model mean differences in electricity production by technology in LAC assuming climate change impacts on renewables.** Differences are calculated by technology using cumulative generation (Terawatt-hours – TWh) during the 2020—2100 period and are relative to the corresponding No-Climate impacts scenarios. LAC regions covered include Brazil, Central America, and the Caribbean (C. Am/Car.), Mexico, South America_Northern (S. Am. (N)), South America_Southern (S. Am. (S)), Argentina, Colombia, and Uruguay (Supplementary Table 4 provides a breakdown of countries per GCAM LAC region). Note the different y-axis scales. Results for the period 2020—2050 are provided in Supplementary Fig. 16.

Second, some LAC subregions (Argentina, C. Am/Car. and Mexico) are characterized by lower present-day and projected contributions of hydropower production compared to others (Supplementary Figs. 3, 5—6, 15). Hence, their national RE portfolios become more sensitive to climate impacts on the non-hydropower renewables. Even in Brazil where climate impacts on hydropower largely govern power-sector responses, important differences concerning wind-based generation exist. Conversely, Colombia and Uruguay are noteworthy cases in which climate impacts on hydropower dominate the effects on the power system as the non-hydropower RE impact signals resulted to be negligible (Supplementary Figs. 11—14).

Focusing on the RCP26 Combined impacts scenarios (columns 5 and 6 in Fig. 1), the magnitudes of changes in wind and solar generation tend to be larger in the NoCCS & NoNewNuc relative to the FullTech, since the former is far more reliant on renewables than the latter (Supplementary Figs. 4—6). Nevertheless, the directions of changes in wind generation in the NoCCS & NoNewNuc are largely consistent with the FullTech (except for where this signal is minor as noted earlier for Colombia and Uruguay), while differences in other non-hydropower sources are mostly indirect effects (i.e., driven by the changes in hydropower and wind; see Supplementary Figs. 11—14). Note that regional responses in hydropower are consistent in scenarios with the same climate forcing because the temporal evolution of hydropower in GCAM follows predetermined input assumptions on

how much hydroelectricity each region will produce per time step. Each climate forcing level (RCP2.6 and RCP6.0) has distinct assumptions (see Methods and Supplementary Note 3 for details on the modeling of hydropower and climate impacts in GCAM). Our results imply that wherever favorable nontrivial signals from the climate-impacted wind resource exist (e.g., Brazil, S. Am. N., and S. Am. S.), wind energy may represent an optimal opportunity to decarbonize the power system, with the potential to also serve as a key adaptation strategy to climate-attributable losses in hydropower (e.g., S. Am. N.). Conversely, Argentina and C. Am/Car. may need to increase generation from a mix of alternative sources to compensate for potential reductions in wind power as the projected positive climate effects on hydropower appear insufficient to satisfy demand.

Figure 1 also emphasizes implications from distinct warming levels. A salient response from the Hydropower scenarios (columns 1, 2, and 3 in Fig. 1) is an overall deterioration of hydroelectricity production under the RCP6.0. All regions, except for Colombia, experience enhanced reductions in cumulative generation, shifts from generation gains toward losses or less pronounced positive impacts compared to the RCP26 Hydropower scenarios. C. Am/Car., Mexico, S. Am. (N) and Argentina emerge as particularly prone to negative impacts on hydropower as the severity of climate change increases. In these regions, a potential adaptation strategy assessed by GCAM might be to increase fossil fuel-based generation (particularly natural gas),

which can exacerbate the initial climate change signal via increments in fossil fuel emissions. A comparison between the RCP60_Baseline: Hydropower and RCP60_Baseline: Combined impacts scenarios (columns 1 and 4 in Fig. 1) reinforces the importance of detailed considerations of multiple impacts, which is particularly prominent in C. Am/Car., Mexico and Argentina. Again, the combination of impacts on hydropower and wind are the leading drivers of the compounding effects on electricity generation, however the direct effects on electricity generation changes induced by the RCP6.0 wind supply curves tend to be less pronounced than those induced by the RCP2.6 curves (Supplementary Figs. 11—14). This is particularly true for Brazil, S. Am. (N), and S. Am. (S). As a result, these regions experience less pronounced gains in wind-based generation under the RCP60_Baseline: Combined impacts relative to the RCP26_Full-Tech: Combined impacts case. It is important to note that these distinct outcomes must not be entirely attributed to the climate change signal due to the role of the energy technology pathway by itself. Specifically, under the RCP60_Baseline scenario, the effects produced by the wind supply curves (shown in Supplementary Figs. 17—19) on wind power generation originate from the lower ends of the curves as wind power needs are not so prominent in this scenario. Conversely, energy-technology pathways like the FullTech and, in particular, the NoCCS & NoNewNuc rely considerably more on wind power to fulfill climate goals, thus suffering stronger influence from upper portions of the supply curves, in which differences among climate-impacted curves are more pronounced.

In all climate-impact scenarios, much of the differences in electricity generation tend to be more pronounced throughout the 2061–2100 period (Supplementary Figs. 29—36). Given the unique implications each subregion may face due to climate impacts on renewables, these results illustrate how distinct accounting of these impacts in IAMs may affect decision-making. For example, under the RCP60_Baseline scenarios, Argentina is projected to experience a pattern of temporally increasing losses in hydroelectricity production (Supplementary Figs. 34–left panels), which would require continuously improving adaptation plans. In this regard, modeling impacts only on hydropower implies that increased wind power generation would be among the portfolio of cost-effective adaptation options in Argentina. On the other hand, accounting for impacts in all renewables means that hydropower losses might be progressively exacerbated by losses in wind power generation, requiring a change in the course of power-sector adaptation plans in the country.

*Implications for power-sector capital investments.* Power-sector capital investments depend on how much generating capacity is installed or retired overtime per technology and the marginal costs of building capacity from each technology (Methods and Supplementary Note 5). Hence, the climate-induced alterations in electricity production patterns discussed so far would have implications for regional capital investment needs through changes in generating capacity (Supplementary Figs. 37—44 compare how our future estimates of generating capacity compare with historical rates). Under the Combined impacts scenarios, our analysis signals increased needs for capital investments in most LAC subregions until 2100, particularly in the NoCCS & NoNewNuc scenario (Fig. 2a). On average, cumulative total capital investment needs in LAC over the 2020—2100 period increase by approximately USD 12–114 billion compared to the No-Climate impacts scenarios (Table 2). Putting these results into context, our highest figure is comparable to LAC's investments in RE accumulated between 2007 and 2015 (of about USD 119 billion), whereas the lowest estimates compare with investments in 2014 or 2015, on the order of USD 15–16

billion[33]. Although these additional investments seem small, they could imply significant challenges for the developing economies in LAC, where resources for public investments are scarcer, and private financing costs (closely linked to perceptions of the quality of institutions and associated investment risks[33,52]) are generally higher compared to the developed world. Among individual subregions, S. Am. (S) stands out with the highest additional investments (of about USD 7–54 billion) in the RCP26 cases. In contrast, investments decrease by USD −0.2 to −5.6 billion in Argentina, Mexico (in the Baseline and FullTech technology cases), and Uruguay.

A breakdown of these total differences by generating source highlights the role of hydropower and wind in altering the net balance of capital investments across LAC (Fig. 3). The regional differences in investments largely reflect the changes to the electricity technology mix shown in Fig. 1. Under the RCP26 Combined impacts scenarios, investments in hydropower and wind-based generating capacity increase in LAC until the end-of-century (greatly influenced by the largest magnitudes of changes in Brazil and S. Am. (S)), while solar- and CCS-based generating capacity lose investments. Nevertheless, important regional variations exist as subregions such as C. Am/Car. and Colombia need to bring solar capacity online. In the RCP60_Baseline: Combined impacts scenario, the net regional investment in hydropower decreases due to the projected negative climate effects on hydropower in many subregions. In this case, the regional increase in total investments is influenced by a net growth in investments in solar energy.

Figures 2 and 3 also illustrate marked differences in capital investments when only climate impacts on hydropower are accounted for. In many regions, such differences translate into underestimated investment needs, which are more pronounced in the RCP26_NoCCS & NoNewNuc case and in Brazil and S. Am. (S.). In these regions, cumulative 2020-2100 capital investment differences in the RCP26_NoCCS & NoNewNuc: Hydropower scenario are approximately USD 60 billion lower than in the RCP26_NoCCS & NoNewNuc: Combined impacts case. Exceptions are Argentina, where reductions in total capital investments in the RCP26 Combined impacts scenarios are considerably larger than in the Hydropower scenarios due to lower wind capacity requirements, and Colombia and Uruguay, where total investment requirements are consistent in both RCP26 climate-impact scenarios because climate impacts on non-hydropower renewables do not play important roles (recall Fig. 1). Under the RCP60_Baseline scenarios, there are also examples in which the Hydropower case do not show lower investment requirements relative to the case of the Combined impact—Mexico and Argentina. However, investment estimates in these subregions under the distinct climate-impact modeling approaches differ markedly.

Although it could be expected that the RCP60_Baseline: Combined impacts scenario would yield considerably larger needs of capital investments in face of more severe climate impacts, we find that investment changes under the RCP60_Baseline: Combined impacts scenario are predominantly lower than or close to those in the RCP26_FullTech: Combined impacts case (Fig. 3 and Table 2). One key aspect is the overall low reliance of the Baseline pathways on RE as pointed out earlier. Under the RCP60_Baseline scenarios, no cost penalties are imposed for emitting fossil fuels, meaning that it is economically attractive to compensate part of renewable-based generation losses by fossil fuels without CCS, typically less capital-intensive than low-carbon options. This dynamic is more evident in Argentina and Mexico. These results then emphasize the role of the energy technology strategy in shaping the overall power-sector vulnerability to climate impacts on RE.

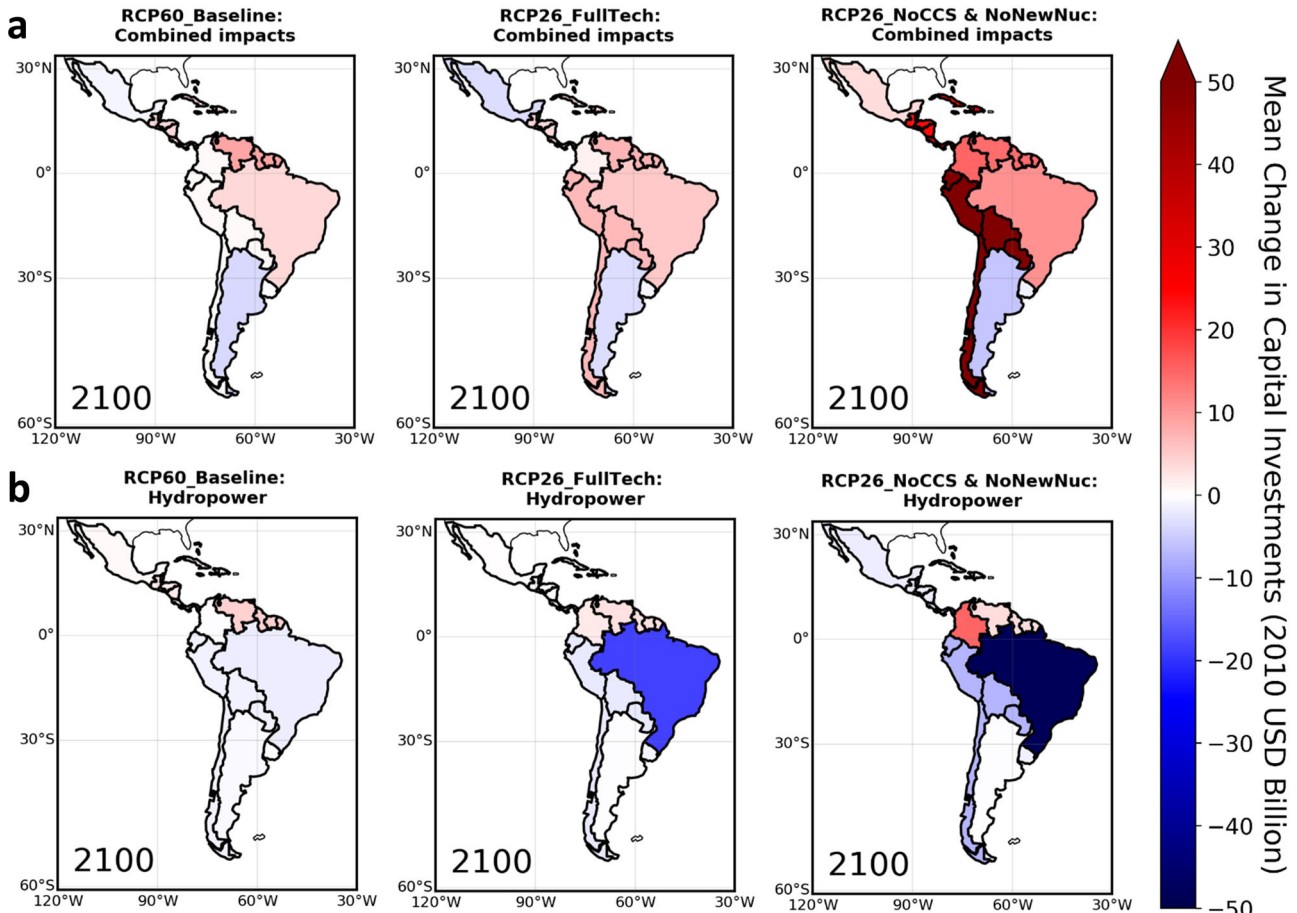

**Fig. 2 Model mean changes in total capital investment requirements in LAC by scenario under distinct assumptions on climate change impacts on renewables.** Absolute differences computed under the Combined impacts scenarios (**a**) and Hydropower scenarios (**b**). Changes are calculated using cumulative capital costs (United States dollar – USD) in the 2020–2100 period and are relative to the No-Climate impacts scenarios (i.e., positive values mean that scenarios with climate impacts on renewables show increased costs). Full range of estimated costs: USD −48 to +54 billion. Results for the period 2020–2050 are provided in Supplementary Fig. 45.

**Table 2 Regionally aggregated changes in total capital investments in the LAC electric power sector under the Combined impacts scenarios.**

| Region | RCP60_Baseline 2100 | | | RCP26_FullTech 2100 | | | RCP26_NoCCS & NoNewNuc 2100 | | |
|---|---|---|---|---|---|---|---|---|---|
| | Mean ($Bill.) | Mean (%) | Std. ($Bill.) | Mean ($Bill.) | Mean (%) | Std. ($Bill.) | Mean ($Bill.) | Mean (%) | Std. ($Bill.) |
| Brazil | 3.72 | 0.42 | 15.54 | 5.32 | 0.30 | 17.84 | 10.76 | 0.48 | 58.74 |
| Central America and Caribbean (C. Am/Car.) | 3.75 | 0.52 | 2.50 | 3.93 | 0.33 | 11.83 | 23.65 | 1.51 | 45.97 |
| Mexico | −0.81 | −0.11 | 3.71 | −3.52 | −0.25 | 16.94 | 3.28 | 0.21 | 17.71 |
| South America_Northern (S. Am. (N)) | 8.71 | 2.59 | 22.98 | 7.07 | 1.22 | 13.54 | 14.09 | 1.99 | 17.55 |
| South America_Southern (S. Am. (S)) | 0.37 | 0.07 | 4.12 | 6.94 | 0.88 | 2.51 | 54.37 | 6.11 | 9.92 |
| Argentina | −3.65 | −1.22 | 1.78 | −3.45 | −0.53 | 1.24 | −5.55 | −0.54 | 0.66 |
| Colombia | 0.48 | 0.19 | 1.49 | 1.28 | 0.25 | 1.61 | 15.13 | 2.05 | 5.87 |
| Uruguay | −0.20 | −0.34 | 0.47 | −0.75 | −0.85 | 0.71 | −1.45 | −1.35 | 0.48 |
| LAC | 12.38 | 0.33 | 46.91 | 16.82 | 0.24 | 37.49 | 114.30 | 1.28 | 129.76 |

Changes represent the mean value (absolute and percentage) across GCMs (the standard deviation of the absolute model mean change is also shown), and are calculated using cumulative investments in the 2020—2100 period. Changes are relative to the No-Climate impacts scenarios (i.e., positive values mean that scenarios with climate impacts on renewables show increased costs). The corresponding results for the period 2020—2050 are provided in Supplementary Table 11.

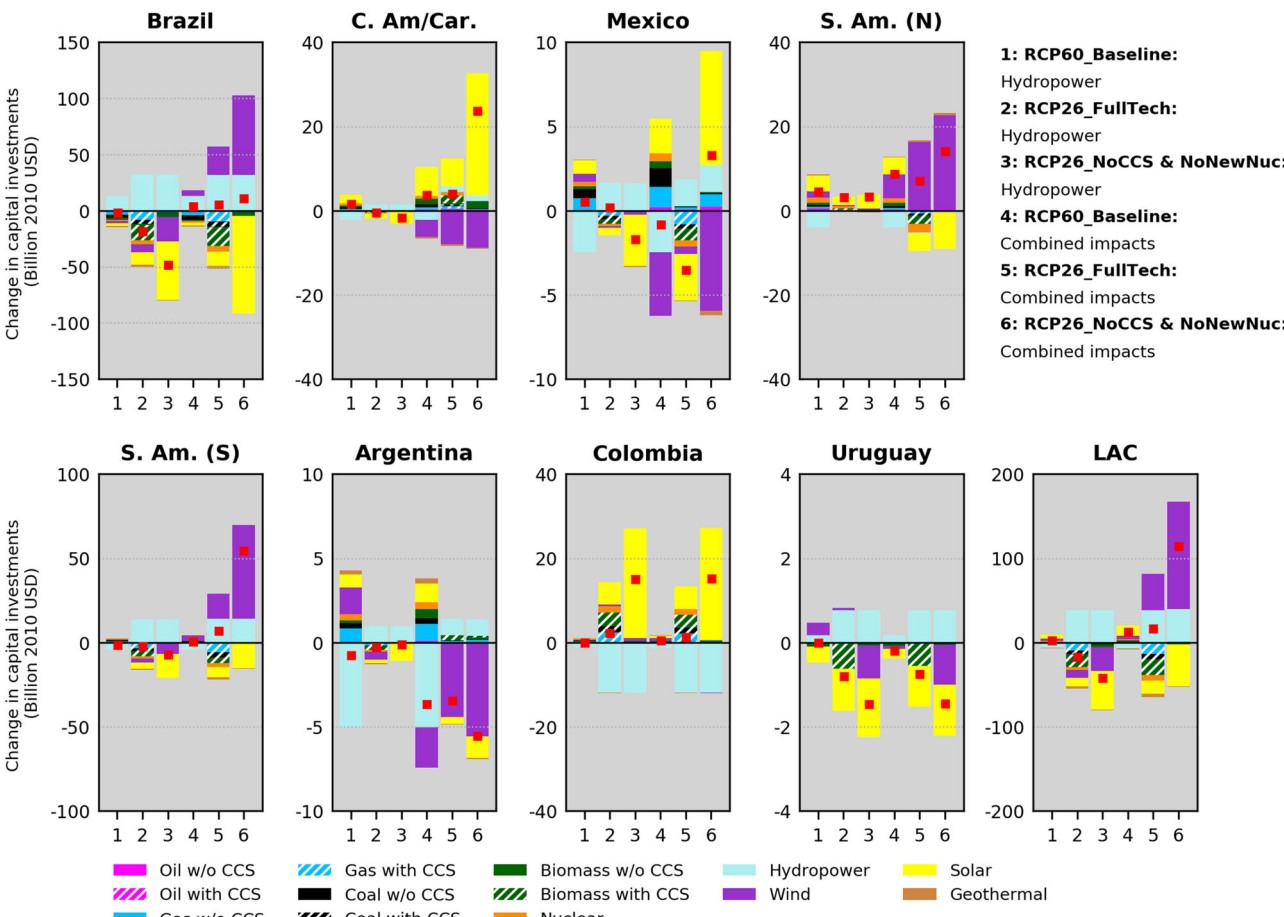

**Fig. 3 Model mean differences in capital investments by technology in LAC assuming climate change impacts on renewables.** Differences are calculated by technology using cumulative investments (USD) in the 2020—2100 period. Differences are relative to the No-Climate impacts scenarios (i.e., positive values mean that scenarios with climate impacts show increased costs). The red squares indicate the net of the positive and negative changes for a given scenario (and are equal to the total investment changes plotted in Fig. 2). Note the different y axis scales. Results for the period 2020—2050 are provided in Supplementary Fig. 46.

We recognize that the investment implications estimated in this analysis are inherently uncertain due to a wide range of outcomes from individual GCM-derived impacts (Supplementary Figs. 47—48). This wide range relates to the substantial uncertainties in GCM projections of variables such as precipitation, winds, and shortwave solar radiation used to force the impact models employed herein. For this reason, uncertainties are high for all technology cases although the NoCCS & NoNewNuc exhibits, for most subregions, the greatest magnitudes of standard deviations associated with the more pronounced mean impacts in this scenario (Table 2). Overall, mean impacts estimated for Brazil, C. Am/Car., Mexico, and S. Am. (N.) are associated with the largest spread of model outcomes (Supplementary Figs. 47—48). Although our ensemble of three climate runs is insufficient to cover the full range of uncertainties across GCMs, it provides initial estimates of overall bounds of economic impacts each region might experience. Importantly, we find larger confidence in investment projections for S. Am. (S), Argentina, Colombia, and Uruguay, particularly under the RCP26 cases, reflected in lower standard deviations (relative to their means) than in other subregions (Table 2) and agreement on the direction of the investment impact (Supplementary Figs. 47—48). Future research should employ a larger ensemble of models to improve overall confidence on the projected changes. Nonetheless, even employing considerably larger ensembles than the one used here, prior studies[20,53] have highlighted the significant decision-making

challenge arising from a large spread of individual model outcomes. To improve the resilience of energy systems in light of the large uncertainty in future climate projections, there are arguments supporting "uncertainty-management" methods[54] like adaptation strategies that are valid under alternative future outcomes, diversify generation sources and consider a more decentralized small-scale energy structure[54–57].

## Discussion

The findings of our study underscore the value of a comprehensive analysis of the implications of climate impacts on RE in IAMs so that their aggregate effect on the energy sector can be better understood. This is important because reductions in total power generation due to climate impacts on one RE source may be alleviated or offset by positive impacts on other sources, or simultaneous negative effects in distinct renewables can amplify total generation losses. GCAM results highlight regionally differentiated impacts across LAC power grids due to a combination of vulnerabilities specific to each generation mix and large spatial variability of climate change impacts across LAC. We explore the first component through distinct technology pathways, showing that the generation portfolio plays an important role in alleviating or exacerbating increasing pressure on capital investments due to climate-attributable effects on renewables. Since each energy technology pathway affects the availability of technology

replacement options (each of them characterized by specific costs of installing generating capacity), implications for total capital investments differ markedly.

The key overarching insight from all scenarios explored herein is the risk of misrepresentation of climate change effects on the electric power sector if climate impacts on all renewables are not accounted for. This is particularly evident for the energy pathway with the most pronounced intermittent renewables deployment (i.e., the NoCCS & NoNewNuc), characterized by greatly underestimated capital investment requirements across most of the LAC region when climate impacts only on hydropower are considered. Such an underestimation may result in enhanced power-sector vulnerabilities to climate change.

To further highlight the importance of a comprehensive analysis, we performed ancillary model experiments assuming climate impacts on each renewable individually (similar to the approach conducted by Turner et al.[20] for hydropower and by Kyle et al.[25] for agricultural yields). These ancillary experiments show that, when all impacts are jointly accounted for in the three Combined impacts scenarios defined earlier, bioenergy and solar generation undergo more pronounced changes than in the experiments where these climate impacts are incorporated individually (Supplementary Figs. 11—14). This results from the compounding price and demand effects of climate impacts on all renewables (see more details on these ancillary experiments in the Supplementary Note 6).

Given the framework of high deployment of intermittent renewables explored through the mitigation scenarios, accounting for climate impacts on wind in certain LAC subregions was shown to be as relevant as accounting for impacts on hydropower in terms of implications for electricity production. Our results also highlighted an overlooked angle related to the fact that climate impacts on wind at the 2 °C warming level can positively affect power production in certain LAC subregions (Brazil, S. Am. (N), and S. Am. (S)). This emerges as a strategic opportunity for decarbonization and diversification of regional power mixes. However, the high upfront capital expenditures of wind technologies (and of renewables in general) represent a critical financial barrier to RE deployment, particularly in developing economies, requiring specific policies to create favorable financing conditions[4,33].

The growing trends in LAC's power-sector capital investment requirements reported under multiple RE impacts and technology configurations suggest challenges for the planning of low-carbon capacity additions. On the one hand, a mitigation pathway based on a diversified mix of generating technologies with sizable contributions from fossil-fueled plants with CCS, as illustrated by the RCP26_FullTech scenario, reduces the exposure of the power system to climate impacts on renewables, and may alleviate (or avoid) the necessity of raising investments. However, CCS technologies are not mature, nor have they been widely deployed commercially yet. On the other hand, decarbonizing LAC's power sector largely through climate-sensitive solar and wind technologies may increase risks of higher capital investment requirements, as shown in Table 2 for most LAC regions under the RCP26_NoCCS & NoNewNuc: Combined impacts scenario. These larger increases relate to the lower capacity factors of intermittent renewables compared with fossil fuels with CCS technologies deployed in RCP26_FullTech: Combined impacts scenario. This means that intermittent renewables require more generating capacity per unit of electricity produced compared with fossil-fuel technologies with CCS (Supplementary Note 5 shows how capacity factors are used to compute capital investments in our methodology). Although the value of diversifying the energy portfolio has been recognized as a means to achieve climate-resilient power systems[55], it is crucial that energy planners identify strategies that do not jeopardize climate goals. In this regard, a mixture of renewable and non-renewable energy sources, albeit less vulnerable to climate impacts on renewables, can dampen mitigation efforts unless CCS technologies become technically viable and cost-competitive and/or comprehensive emissions reduction actions are implemented. Regarding the latter, one alternative might be to focus more heavily on reducing emissions from land and agricultural systems and on enhancing terrestrial sinks for carbon in future decades. This is particularly relevant in LAC where land-related GHG emissions make up a significant share of total emissions[58].

Our analysis is the first to assess the potential implications of climate change impacts on the RE supply for power sector investments in LAC, although our methodology can be used to conduct similar analyses for other regions across the globe. Future studies could also benefit from considering the implications of multiple uncertain factors. One critical aspect noted earlier is the uncertainty originating from the GCMs variables. In addition, hydrological and agricultural yields change assumptions are derived from one impact model each (Methods), however, the structure and parameterization of impacts models are known to be a significant source of uncertainty that can rival that of climate models[59,60]. Another point to note is that our results are focused on aggregated country and regional levels. However, climate change may have distinct and more pronounced effects on smaller sub-national scales. One example is hydropower as climate impacts on runoff patterns are expected to be manifested differently depending on the river basins and sub-basins considered[43]. Hence, further research is needed to develop a finer-resolution multi-impact integrated framework that supports decision-making at sub-national scales. For example, Zarrar et al.[44] contribute to fill such a gap by coupling GCAM and a suite of modeling tools to downscale GCAM projections (part of them including climate impacts on hydropower and agricultural crop yields) onto a grid. This framework was used for a multi-sector assessment of planned policies in Uruguay at a sub-basin scale. Given the possibility of misrepresentation of climate change effects on the power sector highlighted in our results, future high-resolution integrated assessments can benefit from a more comprehensive representation of climate change impacts like the one introduced in this study.

An important caveat of this analysis is that the version of GCAM used in this study represents electricity supply and demand on an annual mean basis assuming, for example, fixed exogenously-defined capacity factors for each power generation technology. Thus, the variability of electricity demand and load at seasonal and daily temporal scales is not considered, which has important implications for decisions on generation infrastructure. The challenge of continuously balancing supply and demand at such finer temporal scales becomes even more complex as the deployment of intermittent solar- and wind-based generation with limited dispatchability increases. Consequently, our analysis likely underestimates rates of capacity additions through 2100 because the annual average supply and demand electricity representation of GCAM smooths out short-term events of peak demand that require the highest electricity outputs. In light of this, our estimates of generation capacity and capital investments should be interpreted as a first-order approximation of the magnitudes of future needs that can be refined by follow-on studies. In this regard, there are ongoing efforts involving GCAM and other IAM groups to improve sub-annual details in power sector representation in IAMs (e.g., Wise et al.[61], Pietzcker et al.[62]). Another consequence of its annual average electricity representation alongside simplifications of important processes is that GCAM cannot represent climate impacts at short timescales (e.g., seasonal scales). These characteristics also impose challenges

for the representation of changes in climate variability and short-term extreme events within IAM frameworks. Hence, our study focuses on implications due to long-term (multi-decadal) mean climatological changes. Future investigation is needed to enhance GCAM modeling capabilities towards finer temporal scales and more detailed representations of power system dynamics. Notwithstanding the limitations above, this study constitutes an additional step toward a more holistic integrated assessment of the potential effects of climate change on the energy sector.

## Methods

**The Global Change Analysis Model (GCAM).** We employ the GCAM[30,63], a global IAM, which maps the interlinkages between human and Earth systems. GCAM is a five-year step dynamic-recursive market-equilibrium model, which is calibrated to a historical base year (2010). The core modeling framework couples: (1) a technology-detailed energy model with representations of supplies and demands; (2) a land and agricultural submodule that provides projections of commodity supply and prices as well as land use and cover changes; (3) a water module that tracks demands in six major sectors; and (4) a reduced-complexity climate model Hector[64].

The driver of demands within the model is the human system (i.e., population and gross domestic product (GDP) growth assumptions), which drives the future evolution of energy, water, and land sectors. On top of socioeconomic trends, a range of mitigation policies, climate impact inputs, adaptation strategies, technological options in distinct sectors, among other assumptions can be added within the scenarios set-up. This allows a multi-sectoral assessment of implications in a way that the model solution represents the least-cost and most technically feasible combination of existing technologies and resources per region. More specifically, given limits imposed by its inputs (costs, current, and future technologies, efficiencies, availability of resources, etc.), GCAM iteratively searches for the set of prices that equilibrates supplies and demands in all sectors. This process aims at finding a solution that minimizes costs or maximizes profits (as in the case of the land sector). However, decision-making in GCAM relies on a logit-choice formulation (Eq. 2), in which preference among competing options depend on their costs (see Eq. 1) or expected profit rates[30]. Although the least-cost or most profitable options capture the largest shares of markets, the other options also gain some market share as explained in the following subsection. Further details on the GCAM are provided in the Supplementary Note 2.

This work was carried out in a research version of GCAM best suited for analyses in LAC (GCAM-LAC)[44], in which important model assumptions have been refined. These include socioeconomic drivers, the disaggregation of Uruguay as a distinct geopolitical region as well as altered parameters related to energy supply, energy demand, and end-use (see Supplementary Table 5 for a list of parameters modified). In GCAM-LAC, the global economy is disaggregated in 33 geopolitical regions, and LAC is represented as eight distinct regions: Argentina, Brazil, Central America and Caribbean, Colombia, Mexico, South America Northern, South America Southern, and Uruguay.

**Climate impacts on renewables—model representation.** Within our impacts modeling framework, GCAM was forced with representations of changing agricultural productivity and hydropower production as well as with climate-impacted solar and wind cost-supply curves. These inputs are based on bias-corrected projections from the GFDL-ESM2M, HadGEM2-ES and IPSL-CM5A-LR general circulation models (GCMs) obtained from the Inter-Sectoral Impact Model Intercomparison Project (ISIMIP)[49,50] under the representative concentration pathways 2.6 (RCP2.6) and 6.0 (RCP6.0). Below we describe how climatic impacts on RE are modeled in GCAM. Further details are provided in the Supplementary Note 3.

To account for climatic impacts on agricultural productivity that affect the modeled biomass production, we used crop yield responses produced by the parallel Decision Support System for Agrotechnology Transfer (pDSSAT - the parallelized global gridded version of the DSSAT model)[65,66] to modify GCAM baseline (i.e., no-climate impacts) crop yield change assumptions (based on the Food and Agriculture Organization projections[25]). The pDSSAT dataset comprises gridded (0.5° spatial resolution) annual yield information for both irrigated and rain-fed crops, which allowed climate-induced yield changes to be applied separately into GCAM rain-fed and irrigated crops. Applying yield estimates from pDSSAT into GCAM requires some data processing to accommodate differences in spatial, temporal, and commodity resolutions between pDSSAT and GCAM (see Supplementary Note 3 for further details on this processing)). One of these key steps is to match crops represented by pDSSAT with the commodities modeled in GCAM. In the specific case of the second-generation bioenergy crops (such as switchgrass, miscanthus, etc), which are not represented by pDSSAT, GCAM's biomass crop commodity receives the median of climate impacts to all other commodities. Note that GCAM requires yield change assumptions to calculate the expected land profitability in each model land unit at each time step. Thus, the effect of the climate-impacted yield change assumptions is to modify such profit rates across land units in the model, which are used to determine land allocated to

each land type (cropland, biomass, grassland, shrubland, pasture, forest, etc.). The combination of yields and endogenous land allocation determines agricultural production in each land unit at each time step[25]. The pDSSAT simulations used in this study are part of the Agricultural Model Inter-comparison Project (AGMIP)[59], and were taken from the experiments that included $CO_2$ effects.

Hydrology simulations from the global hydrological model (GHM) Xanthos[67] were used to modify GCAM default hydropower assumptions, which do not account for climate change impacts. Specifically, hydropower default assumptions (derived from the economic and technical potentials estimated by the International Hydropower Association[30]) are exogenous inputs in GCAM containing predetermined quantities of hydroelectricity production (in EJ) for all time steps and regions. These prescribed quantities that are read in at the start of a simulation then determine the temporal evolution of hydropower production by GCAM region. This means that hydropower production does not result from the modeled economic competition like all other power-sector technologies represented in GCAM.

To incorporate gains/losses in hydropower production under evolving climatic conditions, Xanthos is used to provide information regarding water availability in the 235 large river basins represented in the model, as well as hydropower production To do so, Xanthos requires gridded monthly precipitation and temperature fields from GCMs to solve for monthly runoff and other variables at grid-cell level globally. Using Xanthos 2.0, future projections of hydropower production were computed through a built-in hydropower module (based on ref. [20]) that requires gridded streamflow projections (converted from the simulated runoff) to drive dam simulations. Data processing included aggregation from grid cells to GCAM regions and from monthly to yearly resolution, as well as the smoothing of the resulting yearly hydropower generation pathway to remove inter-annual variability. Lastly, for each GCAM region, absolute hydroelectricity generation (in energy units) in all future years are converted into percent changes relative to 2010 as in turner et al[20]. These percent changes are superimposed onto the default GCAM assumptions producing a modified hydropower production pathway—expressing the amount of hydroelectricity (in EJ) produced in each region at each time step—that incorporates climate change effects.

In this study, we model climate impacts on solar and wind resource productions using exogenous supply curves, which map the availability of energy production as a function of the energy price. These supply curves were built upon the global estimates of renewable energy potentials produced as part of the 'ISIpedia-energy protocol' project[14] using climate variables (e.g., solar radiation, temperature, wind speed) taken from the ISIMIP2b climate simulations. These data consist of gridded (0.5° spatial resolution) maps of technical and economic potentials for four generating technologies (concentrating solar power, photovoltaics—utility-scale and rooftop—and wind), covering three distinct time-slices (1971–2000: historical conditions; 2031–2070 and 2071–2100: future climate states). Detailed methodology behind the derivation of the solar and wind potentials that served as basis for the supply curves implemented in GCAM is documented in Gernaat[29]. Supplementary Note 3 provides a summary of the main steps in such a computation.

To produce supply curves for all GCAM regions, we arranged the technical potential data across the grid cells corresponding to each GCAM region in order of ascending electricity costs (given by the economic potential maps) considering all generating sources, time-periods, and GCMs. This led to the derivation of three time-varying supply curves per renewable source and GCM (Supplementary Figs. 17–28), which we utilized to replace GCAM default assumptions that do not consider climate change effects on the solar and wind primary resource production. In the case of wind, the default supply curves derive from a reanalysis dataset covering the 1980–2009 period[68]. Solar energy is modeled as two separate resources: global solar resource and distributed PV (accounting for PV installations on residential and commercial buildings)[63]. While the distributed PV resource is modeled with supply curves derived from an observational solar radiation dataset[69], no cost-supply curve is implemented for the global solar primary resource (representing utility-scale solar technologies), which is assumed to be an unlimited resource with very low marginal costs that do not vary with deployment levels[63].

Replacing default GCAM assumptions by the modified supply curves implemented in this study has important implications. In GCAM, primary renewable resource production and their marginal resource-related costs serve as inputs to the electricity sector, which contains representations of distinct generating technologies (fossil fuels, geothermal, hydropower, intermittent renewables, and nuclear). The cost of generating electricity given by the renewables supply curves represents the fuel costs that GCAM uses to calculate the levelized cost of the technology $T$ in time period $t$, $p_{T,t}$, given by:

$$p_{T,t} = \frac{C_{\text{fuel}}}{\eta} + \frac{1000\,C_{\text{capital}}}{8760\,\text{CF}} \times \text{FCR} + \frac{C_{\text{O\&M,fixed}}}{8760\,\text{CF}} + C_{\text{O\&M,variable}} \quad (1)$$

where $C_{\text{fuel}}$ is the fuel cost ($ per MWh); $\eta$ is the power plant efficiency; $C_{\text{capital}}$ is the overnight capital cost ($ per kW), CF is the capacity factor of the technology in the investment segment, FCR is the fixed charge rate; $C_{\text{O\&M,fixed}}$ is the annual fixed O&M cost ($ per MW per year); $C_{\text{O\&M,variable}}$ is the variable O&M cost ($ per MWh) and 8760 is the number of hours in a year. The list of electric power generation technologies represented in the model and their input assumptions are documented in Muratori et al[70]. Thus, higher/lower average availability of a

renewable resource due to climate change would translate into shifting supply curves, which would affect $C_{fuel}$ in Eq. 1 above. This would indeed translate into alterations on generating capacity as $p_{T,t}$ is used to compute the share of regional electricity markets each generating technology $T$ captures at time $t$. As mentioned earlier, this market competition is modeled by a logit-choice formulation given by (note that hydropower is set aside from economic competition since hydropower production is a fixed input to the model):

$$s_{T,t} = \frac{\alpha_{T,t} p_{T,t}^{\gamma}}{\sum_{T=1}^{N} \alpha_{T,t} p_{T,t}^{\gamma}} \qquad (2)$$

where $p_{T,t}$ is the levelized cost of the technology $T$ in time period t (Eq. 1) and $\gamma$ is an exogenous input shape parameter. $\alpha_{T,t}$ are calibration parameters (called "share-weights"). This formulation has an important property in that it assigns some market share to expensive technologies, which allows the model to avoid an unrealistic "winner take all" responses based on the notion that choices are based on other factors besides observed prices or that single observed prices do not represent the full variation in prices across applications[61]. Lastly, it is important to mention that GCAM includes a representation of renewable intermittency. Like most IAMs, this is translated into costs that vary with share of renewables in the grid (see further details in Supplementary Note 7).

The impacts on the power system due to climate change on renewables were then examined by comparing scenarios with climate impacts on renewables against identical scenarios that neglect these effects (i.e., the No-climate impacts scenarios) according to the scenarios design presented in Table 1. Note that climate impacts on other relevant aspects of the energy system (e.g., building energy consumption[71], thermal power generation[72,73], transmission infrastructure[14], etc.) were not included in our experimental setup. This means that our results should be interpreted in light of this assumption. Although our modeling framework provides a previously unexplored picture of the effects of climate impacts on all renewables on the power system, future investigation is needed to incorporate impacts on other components of energy system, such as those cited above, which are also acknowledged as key sources of vulnerabilities to the energy system[14].

**Calculation of capital investments in the electric power sector**. Power-sector capital investments calculation follows Iyer et al.[13]. Note that the GCAM representation of capital stock turnover in the electric power sector assumes that generating technologies have a prescribed lifetime, and investments in new plants are added by vintage (i.e., period in which the investment is made) in a pace that allows sufficient generating capacity to satisfy demand. Each power plant operates until the end of its lifetime or is retired from production if its operating costs surpass the electricity market price. The new technology investments compete for a share of energy markets, which is represented by the logit-choice formulation discussed above. Based on GCAM outputs of electricity generation by technology, vintage, and period, we first compute new and additional electricity generation for each technology in each period, which is converted to generating capacity (via capacity factor assumptions). This capacity addition is then converted into capital investments (via overnight capital cost assumptions). Note that the capital investments computed here represent the upfront costs that occur at the beginning of the lifetime of a power station. Variable costs (e.g., fuel costs and operation and maintenance costs) and other system costs (e.g., integration) are not included. Further details and specific assumptions are provided in Supplementary Note 5 and in Iyer et al.[13].

**Reporting summary**. Further information on research design is available in the Nature Research Reporting Summary linked to this article.

## Data availability
The data that support the findings of this study are available at https://github.com/Silviameteoro/GCAM-LAC_Modeling. Source Data are provided with this paper.

## Code availability
The source code of the GCAM-LAC model used in this study is available at https://doi.org/10.5281/zenodo.4048788.

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

## Acknowledgements

We wish to thank the Inter-American Development Bank for sponsoring this effort under contract C0260-16. Additionally, this material is based upon work supported by the National Science Foundation under Grant No. 1855982. The authors are also grateful to the ISIpedia-energy model inter-comparison project, in particular the ISIpedia-NL team, Dr. Detlef P. van Vuuren, Dr. Seleshi G. Yalew, Dr. David E.H.J. Gernaat, Dr. Michelle T.H. van Vliet, and Dr. Fulco Ludwig, for sharing data on climate change impacts on wind and solar supply-curves used in this study.

## Author contributions

S.R.S.d.S., M.I.H., G.I., and F.M.W. designed the research. M.I.H. and G.I. provided guidance on GCAM modeling. S.R.S.d.S., T.B.W, M.B., P.P., A.C.S., and C.R.V carried out modeling in GCAM. S.R.S.d.S. performed the model simulations and conducted data analysis. S.R.S.d.S. wrote the article with major contributions provided by M.I.H., G.I., T. B.W., and F.M.W. All the authors contributed to discussions and to the revision of the paper.

## Competing interests

The authors declare no competing interests.
