## [Peer Review File · Nature Communications]

REVIEWER COMMENTS

Reviewer #1 (Remarks to the Author):

I find this an interesting, generally well written paper that has the potential to contribute well to the field. As far as I know it is novel in its scope. I believe the method is largely appropriate and the paper can make a good contribution in terms of methodology and also policy/system-planning insights.

Overall, the method and results are well presented and clear. However, some clarifications and additional detail are needed to help the reader understand and reproduce the modelling, for example a summary of the new GCAM parameters.

From the main text and the SI, it is clear that the modelling and results have been carefully considered. However, the analysis of the results in the main text seems a bit thin in parts. I recommend that deeper analysis on certain aspects of the scenarios should be worked on to draw out the best value of the paper. For example, deeper analysis of the technology scenarios and climate model uncertainty, perhaps another climate change scenario, and more thorough consideration of the results in the context of national energy systems (current and planned). Then I think the authors would be able to provide important insights on the vulnerability, resilience and investment requirements for electricity systems in the Latin American region.

I have aimed to describe these points in more detail in the comments below.

Introduction

1. Line 36: "Hence, climate change adaptation through alterations in capacity addition plans may be required to balance electricity supply and demand, which could incur in upward pressure on power-sector investments."

This sentence is key for stating the value of the paper and grabbing the reader's attention. I suggest revisiting this when the discussion has been improved (see comments 16, 18, 35-38). For example, it should probably comment on the importance of considering climate change impacts for assessing vulnerability and resilience of the system, so that system planners can ensure energy security at lower costs.

2. Line 39 onwards: This paragraph covers the relevant literature fairly well but it could be better structured. I suggest breaking it up into more distinct points. For example:

- Studies that show climate impacts on the energy system are expected to be significant, particularly in the LAC region.
- Studies that do include climate impacts in IAMs and power sector models
- Description of the research gap – studies are needed that include multiple energy system climate impacts in an IAM, particularly for the LAC region, with focus on impacts on investment requirements.

3. To be thorough, I think there's a few more papers that are relevant, which implement climate impacts in IAMs as bottom-up technical impacts and damage functions, and in power sector models. See those cited and described in

- Cronin et al 2018 <https://doi.org/10.1007/s10584-018-2265-4> (section 5)
- Emodi et al 2019 <https://www.sciencedirect.com/science/article/pii/S0048969719318297>
- Also, Turner 2018 - <https://www.nature.com/articles/s41467-018-07894-4>,
- and Sridharan 2019 <https://pubmed.ncbi.nlm.nih.gov/30655521/>

4. For the modelling of climate impacts, it is important to note the difference between changes in long-term trends, increasing variability and extreme events. Which are you talking about in this literature review and study? If variability and/or extreme events are excluded, justify why, for example because they are thought to be less important or more difficult to model? This should probably be referred to in the discussion as well.

5. Line 63: On the research gap, it is stated that models should include climate impacts on all renewable resources to be comprehensive. I agree with this due to the compound effects as mentioned. However, why is it not also important to include climate impacts on thermal power stations and transmission infrastructure? Some justification for this choice is needed in the lit review and/or method.

6. Line 39: "Most decarbonization scenarios (e.g., those reviewed by the Intergovernmental Panel on Climate Change¹²) suggest that large investments in renewables will be required, particularly under assumptions of limited or no deployment of carbon capture and storage (CCS) and nuclear technologies."

This statement should be backed up by key references that look at supply-side investments in scenarios of low CCS/nuclear deployment.

7. Line 45: "Research on this question has been very limited and the majority of mitigation scenarios in the literature do not account for the impacts of climate change¹²."

Indicate the page where the IPCC report states this research gap

8. Line 60: "Another gap in the literature is the lack of regionally-focused studies."

Reference needed here, for example the IPCC AR5 and Cronin et al 2018.

9. Line 73: "the strategic role of RE in many LAC countries' climate goals"

This is referred to in passing but seems to be key for justifying the importance and relevance of the paper. I recommend some more information on this is needed, in order for the reader to understand the relative importance of the different technologies and climate impacts in the different countries and the technology scenarios. E.g. which countries are planning to use lots more hydropower, which are planning to diversify... etc.

10. Line 76: Specify which figure or note in the SI you are referring to here.

11. The introduction should include some reference to the current technology mix of the LAC countries – or the mix in the reference scenario – so that the reader can understand the importance of each technology, the starting point of climate vulnerability/starting point for the simulations, and the differences between the countries. A brief description would be very useful in the Intro and more description or a chart can be put in the SI.

12. The introduction should also make some reference to previous examination of LAC decarbonisation scenarios, and technologies that may be available in the coming decades, so the reader understands the relevance of the technology scenarios examined in this study.

13. I suggest a few sentences on the climate impacts expected in LAC should be put in the intro main text, with a reference to the SI for more detail.

Experimental Design

14. Line 86: Here, and wherever you refer to the Supplementary Information, please specify which figure or note you want the reader to look at and why

15. Line 92 and 496-514. Please describe in a bit more detail how the RE resources and associated climate impacts are modelled in GCAM.

In particular, please clarify how agricultural productivity and crop yields are related for biomass, and how hydropower availability is defined. E.g. Are these defined with cost supply curves, equivalent to those used for wind and solar? Does hydro availability mean capacity factor here, on what time scale, is it calibrated from historic data, does it include downtime assumptions, is the representation the same for every type/size of hydro power plant, including with dams and run of river, is there any seasonality considered for any of the RE technologies?

Climate impacts are imposed on the cost supply curves for wind and solar, but these are not

mentioned for hydro and biomass. Are hydropower and biomass also implemented as cost-supply curves in GCAM? Either way, describe how the climate impacts affect the parameters which GCAM uses.

16. Line 100 and 216: It is good that the authors have considered climate model uncertainty. I suggest that 3 GCMs is insufficient for a good representation of the climate model ensemble. For example, see comments in the ISI-MIP project about their choice of GCMs, and Carvajal et al 2017 <https://link.springer.com/article/10.1007/s10584-017-2055-4>. Please provide some justification for the choice of GCMs and validity of the method.

Further, the results of the uncertainty assessment do not seem to be considered in detail in the main text or SI. Comment is needed on what the uncertainty assessment shows regarding the robustness of the results and the resilience of the electricity system itself. For example, SI fig 11 shows that for some regions the difference in investments are positive from 1 GCM but negative from another. This uncertainty over the climate impacts would presumably have a significant implication for the extent to which system planners are able to account for climate impacts.

17. Line 101: "Note that the RCP2.6 represents the lowest projected warming level among RCPs, consistent with the long-term goal of the Paris Agreement of keeping global warming likely below 2°C above pre-industrial temperatures¹²."

This statement is out of date as studies are now examining RCP1.9 and looking at scenarios where global warming is limited to 1.5C. Consider revising this sentence to say why you choose RCP2.6 and not higher or lower levels of warming.

18. Scenarios: the descriptions of the scenarios are clear and Table 1 is a useful summary. However, here and in the results below, consider centering the insights from scenario 3 (FullTech: Combined impacts), with description of the other scenarios alongside.

It appears that the purpose of scenarios 1 and 2 (hydropower only) is to show that representation of climate impacts on hydro only is insufficient and gives very different results from when impacts on all RE techs are modelled. This therefore appears to be mainly a modelling contribution of the paper.

In contrast, scenarios 3 and 4 (Combined impacts) provide insights on what could happen in the world – i.e. the climate impact of rcp2.6 and the (un-) availability of CCS and new nuclear. Consider re-focusing the paper on the technology scenarios with combined impacts, climate model uncertainty, and perhaps a scenario of stronger climate change for comparison (e.g. RCP4 or 6).

Implications for electricity generation patterns

19. Line 115. Please add some description of the changes observed in scenario 1 or 3 relative to the ref scenario. I believe this is needed to help orientate the reader's understanding of the results.

Alternatively, restructure the results to centre the insights from scenario 3, with description of the other scenarios alongside, as described in comment 18.

In any case, the structure of this section should be adjusted to clarify the impact of each scenario change i.e. to highlight the effects of modelling multiple climate impacts, the effects of climate change itself, the effects of the technology scenarios.

20. I suggest the results should describe key elements of the energy system technology transition, as well as the cumulative generation, as the timing of these changes is presumably very relevant for the investment requirements/costs.

21. Line 120: The terms 'non-trivial' and 'negligible' response are used throughout the paper. This should be defined if it implies a specific judgement on which responses are trivial and non-trivial.

22. Fig 1: The clarity of this figure is appreciated. Some information should be included to describe the starting technology mix of each country, or the mix in the ref scenario. As mentioned above, some more information on this is needed in the intro and SI.

23. Line 146: "Note that the responses in hydroelectricity are consistent in all scenarios because the temporal evolution of hydroelectricity production per GCAM region is exogenously predetermined 25 (i.e., fixed for scenarios with and without impacts on hydropower)."

Clarification needed – I understand this sentence to mean that hydropower generation is not, and cannot be, impacted by climate change in the modelling. However, I see this can't be right, as the rest of the paper explains a climate change impact on hydro generation is modelled, and Fig 1 shows there is an impact on hydro generation in all the scenarios relative to the Ref scenario. Please adjust this to explain why the climate impact on hydropower is the same in all scenarios relative to the Ref. i.e. please clarify the meaning of "the temporal evolution of hydroelectricity production is exogenously predetermined." Fixed to what levels and why?

Implications for power-sector capital investments

24. Line 163: On average, total capital investment needs in LAC increase by approximately USD 17–114 billion compared to the No-Climate impacts scenarios (Table 2). State here what these costs represent (cumulative investments over the period 2020 -2100?).

25. Line 168. This is an interesting finding, and the context and comments in this paragraph are very useful. The investments do seem small compared to the recent historic investments in the region. I recommend moving this comparison to the discussion and commenting on whether this indicates climate change impacts should be accounted for by policy makers/system planners, or if they are of little concern. Consider the uncertainty on the results alongside this.

26. Line 185: "The regional patterns in additional investments largely follow the effects induced by the climate-impacted RE inputs on electricity production outlined in Fig. 1"

This sentence is a bit unclear. Consider rephrasing it to something like "The country-level additional investment results largely reflect the changes to the electricity technology mix shown in Fig 1."

Furthermore, is the headline message that where climate change increases the primary resource, the model indicates that the least cost option is to install more of that technology so investments are directed towards that technology, and vice versa? If so, I recommend adding a sentence to this effect.

27. Line 191: Ensure all region names are consistent throughout the paper. E.g. C. Am. C. or C. Am/Car

28. Line 210: "the negligible role of climate impacts"

Clarify if this means the climate impacts on non-hydro renewables are small, or if non-hydro renewables account for a small portion of the electricity generation mix

29. Table 2 Caption: Clarify if the total capital investments include all capital investments for the energy system, or just the electricity sector, or just RE technologies, or just utility scale RE etc.

Discussion

30. Line 248: I understand this sentence to mean that more wind capacity (incentivized by the positive impact of climate change on the wind resource) implies higher investments in wind, which I think is self-evident. Presumably the higher wind capacity displaces one or more other technologies. So here, the total capital investments should be compared rather than the investments in just wind.

31. Line 250: "In this context, Brazil (which has led wind energy expansion in LAC) and Chile (part of S. Am. (S)) are currently more advanced in creating favorable institutional frameworks for investments in renewables than other countries within S. Am. (S) and S. Am. (N)28."

It is unclear how this is related to the previous statement. Discussion of how the results of these

study compare to historic rates of capacity expansion, and the relevance of institutional frameworks would be extremely valuable and require more space devoted to them.

32. Line 256: "diversified mix of generating technologies with sizable contributions from CCS and, thus, lower exposure to climate impacts on renewables,"

Check the rigour of this sentence. Do you mean CCS on fossil fueled thermal power stations? If so, it is the continued use of fossil fuels, facilitated by CCS, which reduces the exposure of the system to climate impacts on renewables.

To conclude that a diversified mix of renewables reduces the exposure to climate impacts, you would need to examine the correlation of climate impacts between different RE technologies in each country, analysis of which is not prominent in this paper.

33. Line 260: "decarbonizing LAC's power sector largely through climate-sensitive solar and wind technologies may increase risks of higher capital investment requirements. "

This should be tied back to the results of this study, e.g. 'as was demonstrated in the comparison of X and Y scenarios for A/b/c countries."

34. Line 262: "One key point is the significantly lower capacity factors of intermittent renewables compared with other technologies, which means that both RE sources require more capacity to generate the same amount of electricity than other technologies, such as fossil fuels with CCS. "

This is technically true but not highly relevant, as capacity is not fully equivalent for different technologies – it is more interesting to discuss total system investment costs or the use of land or natural resources, or efficiency of the system.

35. There is little description of the impact of the two technology scenarios. These are a very interesting contribution of this paper so should be highlighted and discussed in more detail.

36. The interplay between elements of the electricity system should be explored further. i.e. which technologies displace others in each of the scenarios, and how do the climate impacts combine?

37. The authors constructed the scenarios and modelling to examine the impact on the energy system if each of the climate impacts is modelled separately then in combination. This is very interesting but is not discussed in much detail. What can be deduced from this, in terms of the vulnerability of electricity systems which are more or less diversified in a changing climate?

38. Finally, the implications for the electricity systems in these countries should be discussed. What are the key messages for the system planners? Can the results be discussed in terms of whether they indicate certain vulnerabilities, or opportunities to increase resilience? And the cost of mitigating the risks?

39. Line 262 – 276. These are interesting but quite general statements about renewables and energy systems, without direct relevance to the scope of this study. These should be removed to make space for more in depth discussion on the technology scenarios, interplay between elements of the electricity system and wider implications of the investments results.

Methods

40. Line 474: I suggest this should say 'socio-economic drivers' or parameters or trends

41. Line 477: Please clarify this sentence. In what ways does GCAM find the 'most technically feasible combination of technologies...?'

Is it in fact the least cost combination, subject to exogenous constraints which represent the technically feasible costs and availabilities of primary energy resources and supply and demand side technologies?

42. Line 482: I suggest this should say 'socio-economic drivers' or parameters or trends

43. Line 482: I suggest this should say "the disaggregation of Uruguay" because "break-up"

sounds like Uruguay itself has been split up.

44. Line 482: For the sake of being re-producible, I suggest some of these improved parameters should be summarized in the SI.

45. Line 492: Specify which part of the SI

Supplementary Information

46. SI Figure 1 - I am unclear on the meaning of these two charts. Please clarify the legend. What does 'share of RE ...relative to the rest of the world' mean? I think the share of RE in a region's power generation system should be in terms of GWh of RE generation/GWh total generation...?

I hope these comments are helpful and am very happy to discuss further and review further versions.

Jen Cronin

Reviewer #2 (Remarks to the Author):

This is a very interesting paper with a lot of ambition and could provide some good insights into the effects of climate change on renewable energy resources, and could potentially be useful for long-term energy planning exercises. The work explored in the paper could also stimulate follow-up work in more micro (national) studies. There are some minor areas for the authors to carry out their adjustments and/or provide their responses:

1. The authors discuss the existence of literature around the climate change impacts on hydropower; but there is little about the climate change impacts for solar and wind resources. The question then is how do we deal with the high level of uncertainty associated with solar and wind? A bit more discussion is needed here.

2. Why only use the RCP 2.6? It may be that you would like to be consistent with the Paris Agreement, but what happens in the possible chance of higher temperatures. Presumably this would mean different level of impact on renewable resources. Why did you lock yourselves to the stringent RCP 2.6? It would have been very valuable to also look at a less stringent pathway (RCP), which is not only possible, but also may offer additional analysis to decision makers that delayed action on climate change may lead to heavier renewable energy costs in the future.

3. How were the decisions for the scenarios made? and how were the calculations for the estimates done? And what data were used?

4. At the end of the discussion, the authors make the point that 'finer resolution, multi-impact integrated framework' studies would be needed to support national level decision makers. It would be useful to say a bit more on this to provide some guidance on what such national studies can gain from your study and how they can build on it.

Reviewer #3 (Remarks to the Author):

Review on

'Power sector investment implications of climate impacts on renewable resources in Latin America and the Caribbean'

The paper assesses the impact of climate induced changes to renewable energies on the overall

energy system capacity.

Overall, I consider the paper clearly structured and well written and the results add to the overall energy picture on climate change impacts.

Being more familiar with electricity modeling, I nevertheless have one fundamental concern with the approach and resulting interpretation. If I understand the model approach correctly the generation mix is based on costs structures accounting for the fact that renewables have a lower overall output due to intermittency. However, no detailed time structure on demand or supply sides is included (i.e. the capacity is defined by the total output and the capacity factor of each technology).

While such an approach is well suited for dispatchable technologies it does not really account for intermittent wind and solar generation. As those are central for future electricity systems their specifications need to be accounted when making future assessments.

The main problem arising from an average (per year) approach is the simple fact that weather fluctuations (= the intermittency of wind and solar) as well as fluctuations between years (windy vs. less wind years) will require a different capacity approach to keep the electricity system stable and working.

Let's make an example: if there would be an unlimited zero cost solar PV option in your model, that should be the first and only technology to be installed (up to the point where total demand = total generation of this PV technology) if I understand the model correctly. If this is the case than its obvious that the model has a shortcoming, as such a system would not provide sufficient electricity at all times.

This general problem will lead to the requirement to either install significant overcapacity of RES or a complete back-up system of conventional units.

In the first case, the presented changes in overall output due to climate change may not have a feedback on the overall capacity investments at all. In such a world they are way higher than the capacity formula in the methods section ($cap=gen/capfac$) would imply; simply to be able to account for weather variations, seasonality and other uncertainties. This would mean that the higher/lower average availability due to climate change would not necessarily translate into alterations on the capacity side at all.

In the second case, there is a stable back-up system that would need to be built up regardless of the actual average RES injection as such a system would need to be designed for the no-RES-infeed hours anyway (as long as they don't change due to climate change there is no feedback even if a higher total output of RES in a year would be possible). Thus whether altered RES injection would lead to altered RES capacity investments is not so clear.

I understand that this is nothing that the underlying model can easily include and is likely also not the objective of the paper. The point of altered RES potential and feedback to the overall energy system is valid nonetheless. But it would still require a significant rewrite of the discussion to properly account for this problem.

And maybe the capacity part should be skipped completely and the focus should solely be on TWh numbers. It simply is not really possible to derive capacity projections for electricity systems with high shares of intermittent RES without going into a finer time resolution. And capacity projections based on yearly averages are way off by design.

Best regards
Hannes Weigt

Responses to Reviewer 1:

Reviewer #1 (Remarks to the Author):

I find this an interesting, generally well written paper that has the potential to contribute well to the field. As far as I know it is novel in its scope. I believe the method is largely appropriate and the paper can make a good contribution in terms of methodology and also policy/system-planning insights.

Overall, the method and results are well presented and clear. However, some clarifications and additional detail are needed to help the reader understand and reproduce the modelling, for example a summary of the new GCAM parameters.

From the main text and the SI, it is clear that the modelling and results have been carefully considered. However, the analysis of the results in the main text seems a bit thin in parts. I recommend that deeper analysis on certain aspects of the scenarios should be worked on to draw out the best value of the paper. For example, deeper analysis of the technology scenarios and climate model uncertainty, perhaps another climate change scenario, and more thorough consideration of the results in the context of national energy systems (current and planned). Then I think the authors would be able to provide important insights on the vulnerability, resilience and investment requirements for electricity systems in the Latin American region.

Thank you for the positive comments and very constructive suggestions. We have detailed below the revisions and additions to the manuscript in response to them.

I have aimed to describe these points in more detail in the comments below.

1. Line 36: “Hence, climate change adaptation through alterations in capacity addition plans may be required to balance electricity supply and demand, which could incur in upward pressure on power-sector investments.”

This sentence is key for stating the value of the paper and grabbing the reader’s attention. I suggest revisiting this when the discussion has been improved (see comments 16, 18, 35-38). For example, it should probably comment on the importance of considering climate change impacts for assessing vulnerability and resilience of the system, so that system planners can ensure energy security at lower costs.

We have revised the sentence as suggested. Revised sentence now reads:

“Hence, planners need to account for climatic impacts on RE during capacity development planning to ensure power system reliability, which is particularly relevant in the context of decarbonization strategies centered on RE expansion.”

2. Line 39 onwards: This paragraph covers the relevant literature fairly well but it could be better structured. I suggest breaking it up into more distinct points. For example:

- Studies that show climate impacts on the energy system are expected to be significant, particularly in the LAC region.
- Studies that do include climate impacts in IAMs and power sector models
- Description of the research gap – studies are needed that include multiple energy system

climate impacts in an IAM, particularly for the LAC region, with focus on impacts on investment requirements.

We have organized our literature review in the order suggested. A note here is that we have opted for keeping specific literature on climate impacts in the LAC region in the last paragraph of the “introduction” section to better address reviewer’s comment #13. Due to the rearrangements made, we have divided the original paragraph into two paragraphs (see additional comments in our response to reviewer’s comment #3). The two revised paragraphs now read:

“Most decarbonization scenarios (e.g., those reviewed by the Intergovernmental Panel on Climate Change (IPCC)¹²) suggest that large investments in renewables will be required, particularly under assumptions of limited or no deployment of carbon capture and storage (CCS) and nuclear technologies¹³. In this context, there is an open question about how climate impacts on renewable resources – such as those described above – could alter the understanding of the economic implications and investment needs suggested by alternative decarbonization pathways. Research on this question has been very limited and the majority of mitigation scenarios in the literature do not account for the impacts of climate change. This is the case of the about 900 mitigation scenarios reviewed in the IPCC’s Fifth Assessment Report (AR5)¹². Even the few studies exploring climate impacts within the context of decarbonization scenarios have focused only on hydropower without a comprehensive analysis of impacts on all renewable sources.

With growing literature highlighting that the energy sector, including RE production, may face serious impacts due to climate change¹⁴⁻¹⁶, there have been efforts to incorporate climate impacts on renewables into energy and integrated assessment models (IAMs) to support decision-making. Methodologically, many of these studies rely on detailed process-based models (for example, hydrologic models, crop models, general circulation models (GCMs)) capable of simulating climate-impacted environmental responses that are used to modify IAM parameters linked to RE production. However, hydropower – the renewable that currently contributes the most to the global electricity supply⁴ – has received considerably larger attention from the IAM literature and climate-impact studies in general (see the literature referenced in Yalew et al.¹⁴, Solaun and Cerdá¹⁵, Cronin et al.¹⁶, and Emodi et al.¹⁷). IAM-based studies on climate impacts on hydropower (some of them conducted in the context of decarbonization scenarios as mentioned earlier) have been useful in exploring climate change implications for electricity production and capital investments¹⁸⁻²⁴. Another group of IAM-based studies has addressed impacts on the agriculture sector (which affect biomass potential) by incorporating biophysical crop yield changes²⁵⁻²⁸. Regarding the representation of climate impacts on solar and wind resources in IAMs, research efforts are still incipient, and to the best of our knowledge, limited to only two studies^{24,29}. Consequently, there is a gap in the literature on a comprehensive analysis of climate impacts on all renewable resources and their implications for electricity sector investments. Studies that focus on climate impacts on individual resources do not account for the compounding effects of climate impacts on multiple renewable sources and may thus under- or over- estimate investment requirements. Another gap in the literature is the lack of regionally-focused studies^{15,16}. While global studies are useful in characterizing the scale of a problem, policy decisions are made at national to sub-national scales. Hence, regional analyses with focus on national issues and circumstances are important to enhance relevance of the analyses to decision-makers. Our study fills both of the above gaps.”

3. To be thorough, I think there's a few more papers that are relevant, which implement climate impacts in IAMs as bottom-up technical impacts and damage functions, and in power sector models. See those cited and described in

- Cronin et al 2018 <https://doi.org/10.1007/s10584-018-2265-4> (section 5)
- Emodi et al 2019 <https://www.sciencedirect.com/science/article/pii/S0048969719318297>
- Also, Turner 2018 - <https://www.nature.com/articles/s41467-018-07894-4>,
- and Sridharan 2019 <https://pubmed.ncbi.nlm.nih.gov/30655521/>

Thank you for the suggestions on literature. We agree on the relevance of many papers included in the material suggested. Given the vast literature covering specifically climate impacts on hydropower (including their implementation in IAMs and power sector models), and to the fact that we are on the limit of number of references allowed to be included in the article (70 according to the journal guidelines), we have opted for referencing readers to recent literature reviews on climate impacts on the energy sector (Yalew et al 2020, Solaun and Cerdá 2019, Cronin et al 2018, and Emodi et al 2019). This way, we believe interested readers can find a large sample of relevant literature. We have not found new references beyond what we had originally on the implementation of climate impacts on non-hydropower renewables in IAMs or power sector models. Hence, no new references regarding non-hydropower renewables were included.

References:

- Yalew, S. G. *et al.* Impacts of climate change on energy systems in global and regional scenarios. *Nature Energy*, doi:10.1038/s41560-020-0664-z (2020).
- Solaun, K. & Cerdá, E. Climate change impacts on renewable energy generation. A review of quantitative projections. *Renewable and Sustainable Energy Reviews* **116**, 109415, doi:<https://doi.org/10.1016/j.rser.2019.109415> (2019).
- Cronin, J., Anandarajah, G. & Dessens, O. Climate change impacts on the energy system: a review of trends and gaps. *Climatic Change* **151**, 79-93, doi:10.1007/s10584-018-2265-4 (2018).
- Emodi, N. V., Chaiechi, T. & Beg, A. B. M. R. A. The impact of climate variability and change on the energy system: A systematic scoping review. *Science of The Total Environment* **676**, 545-563, doi:<https://doi.org/10.1016/j.scitotenv.2019.04.294> (2019).

4. For the modelling of climate impacts, it is important to note the difference between changes in long-term trends, increasing variability and extreme events. Which are you talking about in this literature review and study? If variability and/or extreme events are excluded, justify why, for example because they are thought to be less important or more difficult to model? This should probably be referred to in the discussion as well.

We have focused on implications to the power system owing to long-term (multi-decadal) mean climatological changes. This relates to the modeling structure of GCAM as explained in the revised last paragraph of the “Discussion” section (note that part of this discussion was originally in the Methods section “Climate impacts on renewables — model representation,” which was removed to avoid repetition). We also note that representing climate variability and climate extreme events in IAMs is currently a very challenging problem, which will require further research efforts across the IAM community.

New text included in the last paragraph of the “Discussion” section reads:

“An important caveat of this analysis is that the version of GCAM used in this study represents electricity supply and demand on an annual mean basis assuming, for example, fixed exogenously-defined capacity factors for each power generation technology. Thus, the variability of electricity demand and load at seasonal and daily temporal scales is not considered, which has important implications for decisions on generation infrastructure. The challenge of continuously balancing supply and demand at such finer temporal scales becomes even more complex as the deployment of intermittent solar- and wind-based generation with limited dispatchability increases. Consequently, our analysis likely underestimates rates of capacity additions through 2100 because the annual average supply and demand electricity representation of GCAM smooths out short-term events of peak demand that require the highest electricity outputs. In light of this, our estimates of generation capacity and capital investments should be interpreted as a first-order approximation of the magnitudes of future needs that can be refined by follow-on studies. In this regard, there are ongoing efforts involving GCAM and other IAM groups to improve sub-annual details in power sector representation in IAMs (e.g., Wise et al.⁶⁴, Pietzcker et al.⁶⁵). Another consequence of its annual average electricity representation alongside simplifications of important processes is that GCAM cannot represent climate impacts at short timescales (e.g., seasonal scales). These characteristics also impose challenges for the representation of changes in climate variability and short-term extreme events within IAM frameworks. Hence, our study focuses on implications due to long-term (multi-decadal) mean climatological changes. Future investigation is needed to enhance GCAM modeling capabilities towards finer temporal scales and more detailed representations of power system dynamics.”

5. Line 63: On the research gap, it is stated that models should include climate impacts on all renewable resources to be comprehensive. I agree with this due to the compound effects as mentioned. However, why is it not also important to include climate impacts on thermal power stations and transmission infrastructure? Some justification for this choice is needed in the lit review and/or method.

We intend to make the point of the need of a representation of climate impacts on all renewables in IAMs by contrasting this approach with a representation of impacts only in hydropower (the focus of numerous prior IAM studies). However, we agree that a comprehensive representation of climate impacts on the energy sector in IAMs must include, apart from climate impacts on the renewable supply, impacts on other relevant components of the energy system. We have updated the “Methods” section (“Climate impacts on renewables — model representation”) to discuss the need of further investigation that accounts for climate impacts on other key energy system components as well. (Note that we have deleted some sentences related to this discussion that were originally in the “Discussion” section to avoid repetition).

New text included in the Methods section: Climate impacts on renewables — model representation (last paragraph) reads:

“Note that climate impacts on other relevant aspects of the energy system (e.g., building energy consumption⁶¹, thermal power generation^{62,63}, transmission infrastructure¹⁴, etc.) were not included in our experimental setup. This means that our results should be interpreted in light of this assumption. Although our modeling framework provides a previously unexplored picture of

the effects of climate impacts on all renewables on the power system, future investigation is needed to incorporate impacts on other components of energy system, such as those cited above, which are also acknowledged as key sources of vulnerabilities to the energy system¹⁴.”

6. Line 39: “Most decarbonization scenarios (e.g., those reviewed by the Intergovernmental Panel on Climate Change¹²) suggest that large investments in renewables will be required, particularly under assumptions of limited or no deployment of carbon capture and storage (CCS) and nuclear technologies.” This statement should be backed up by key references that look at supply-side investments in scenarios of low CCS/nuclear deployment.

Although we have identified a group of peer-reviewed studies exploring technology availability that included mitigation scenarios of low CCS/nuclear deployment, we could only find one study (Iyer *et al* 2017) looking specifically into power-sector capital investments under such circumstances. The study by Iyer *et al* 2017 (reference #13) was included as a reference for this statement.

Reference:

- Iyer, G. et al. Measuring progress from nationally determined contributions to mid-century strategies. *Nature Climate Change* 7, 871-874, doi:10.1038/s41558-017-0005-9 (2017).

7. Line 45: “Research on this question has been very limited and the majority of mitigation scenarios in the literature do not account for the impacts of climate change¹².” Indicate the page where the IPCC report states this research gap

The IPCC AR5 does not explicitly state this research gap, to the best of our knowledge. We have referred readers to the IPCC Synthesis Report (reference #12 in the manuscript) because it represents an example of a very large sample of mitigation scenarios (about 900 as noted on page 81) published in the peer-reviewed literature without climate impacts being accounted for. We have revised this sentence in order not to imply that this research gap is explicitly stated in the IPCC AR5. Revised text now reads:

“Research on this question has been very limited and the majority of mitigation scenarios in the literature do not account for the impacts of climate change. This is the case of the about 900 mitigation scenarios reviewed in the IPCC’s Fifth Assessment Report (AR5)¹².”

8. Line 60: “Another gap in the literature is the lack of regionally-focused studies.” Reference needed here, for example the IPCC AR5 and Cronin et al 2018.

We have included as references for this statement the studies by Solaun and Cerdá 2019 and Cronin *et al* 2018 (references #15 and 16).

References:

- Solaun, K. & Cerdá, E. Climate change impacts on renewable energy generation. A review of quantitative projections. *Renewable and Sustainable Energy Reviews* **116**, 109415, doi:<https://doi.org/10.1016/j.rser.2019.109415> (2019).
- Cronin, J., Anandarajah, G. & Dessens, O. Climate change impacts on the energy system: a review of trends and gaps. *Climatic Change* **151**, 79-93, doi:10.1007/s10584-018-2265-4 (2018).

9. Line 73: “the strategic role of RE in many LAC countries’ climate goals” This is referred to in passing but seems to be key for justifying the importance and relevance of the paper. I recommend some more information on this is needed, in order for the reader to understand the relative importance of the different technologies and climate impacts in the different countries and the technology scenarios. E.g. which countries are planning to use lots more hydropower, which are planning to diversify... etc.

We have included a new table in the Supplementary Information (currently Supplementary Table 1), in which we have summarized the plans regarding renewable energy deployment included in LAC countries’ nationally determined contributions and long-term strategies submitted to the UNFCCC. Given the amount of information consolidated in Supplementary Table 1, we have opted for briefly describing plans of the major LAC economies in the main text. (Note that we are not showing Supplementary Table 1 here because it is a very long table.) New text reads:

“This growth is expected to continue due to strong policies³³, and the strategic role of RE in many LAC countries’ climate goals. As summarized in Supplementary Table 1, 25 LAC countries intend to foster RE as a route for mitigation. Among major economies, Brazil plans to promote non-hydropower sources, Mexico intends to focus on wind, solar photovoltaics (including distributed systems) and hydropower, and Argentina is particularly interested in promoting biofuels.”

10. Line 76: Specify which figure or note in the SI you are referring to here.

Reference is to Supplementary Note 1. We have revised the text to refer explicitly to Supplementary Note 1.

11. The introduction should include some reference to the current technology mix of the LAC countries – or the mix in the reference scenario – so that the reader can understand the importance of each technology, the starting point of climate vulnerability/starting point for the simulations, and the differences between the countries. A brief description would be very useful in the Intro and more description or a chart can be put in the SI.

We have included in the “Introduction” section an overview of current energy technology mix in LAC countries and two additional figures in the Supplementary Information (Supplementary Figs. 2 and 3) to complement this discussion. New text added in the “Introduction” section reads:

“Despite the increasingly important role of RE in LAC, notably in the electricity sector, current regional consumption of fossil fuels remains a challenge for climate mitigation (fossil fuels represented roughly 70% of total primary energy supply (TPES) in 2015³⁴). TPES in LAC depends primarily on oil, natural gas, bioenergy, and hydropower³⁴ with large oil and biofuels use in the transportation sector, while hydropower, natural gas and oil comprise most of the

electricity supply (Supplementary Fig. 2). However, at sub-regional/country levels, important departures from the overall LAC profile exist. For example, regarding hydropower, which dominates regional electricity mixes (notably in Uruguay, Brazil and Colombia), except for Mexico, Central America and the Caribbean and Argentina where main generating sources are natural gas and oil (Supplementary Fig. 3).

12. The introduction should also make some reference to previous examination of LAC decarbonisation scenarios, and technologies that may be available in the coming decades, so the reader understands the relevance of the technology scenarios examined in this study.

In the revised “Introduction” section, we have referred to prior decarbonization scenarios developed for LAC and technologies expected to play key roles under climate policy (renewables, particularly biomass, solar and wind, and CCS technologies applied to fossil fuels and biomass). New text added in the “Introduction” section reads:

“Absent efforts to constrain emissions, fossil technologies in LAC are projected to expand³⁵ (Supplementary Figs. 2–3 provide projections from the GCAM *Baseline (No Policy)* scenario, which assumes no emissions mitigation actions throughout the 21st century). In light of this, previous LAC decarbonization scenarios agree that renewables, mainly biomass, solar and wind as well as CCS technologies applied to fossil fuels and biomass are critical to mitigate energy-sector emissions, with nuclear energy typically playing less relevant roles³⁶⁻⁴¹. In these scenarios, hydropower remains important, but its contribution to regional total generation falls over time, as hydropower capacity expansion is not expected to follow growing demands¹⁵.”

13. I suggest a few sentences on the climate impacts expected in LAC should be put in the intro main text, with a reference to the SI for more detail.

Thank you for the suggestion. New text added in the “Introduction” section (last paragraph) reads:

“Under future climate change conditions, RE production in LAC will potentially face several challenges. By the end of the 21st century, multi-model projections using the representative concentration pathways (RCPs)¹² show mean warming levels reaching 0.6°C to 2.0°C in RCP2.6 and 2.2°C to 7.0°C in RCP8.5, and both positive and negative rainfall anomalies across the region⁴². Although there is large uncertainty intrinsic to these climate projections, their effect on future estimates of hydropower potential is manifested in terms of a strong regional variability of impacts from gains in Uruguay and the southernmost basins of Brazil to losses in northern Brazil, Colombia, northern South America, Argentina, and southern South America^{20,43-45}. The limited literature focusing on LAC suggests increased wind and solar resource potentials in Brazil^{43,46-48}, and, possibly, a positive general response of the main LAC bioenergy feedstock, sugarcane, to regional climatic changes⁴². For more details see Supplementary Note 1 (which briefly reviews main climate-attributable effects on RE alongside anticipated impacts for LAC).

Experimental Design

14. Line 86: Here, and wherever you refer to the Supplementary Information, please specify which figure or note you want the reader to look at and why.

The entire manuscript has been revised and updated to specify which figure or note is being referred to in the main text.

15. Line 92 and 496-514. Please describe in a bit more detail how the RE resources and associated climate impacts are modelled in GCAM.

In particular, please clarify how agricultural productivity and crop yields are related for biomass, and how hydropower availability is defined. E.g. Are these defined with cost supply curves, equivalent to those used for wind and solar? Does hydro availability mean capacity factor here, on what time scale, is it calibrated from historic data, does it include downtime assumptions, is the representation the same for every type/size of hydro power plant, including with dams and run of river, is there any seasonality considered for any of the RE technologies?

Climate impacts are imposed on the cost supply curves for wind and solar, but these are not mentioned for hydro and biomass. Are hydropower and biomass also implemented as cost-supply curves in GCAM? Either way, describe how the climate impacts affect the parameters which GCAM uses.

No, hydropower and biomass are not represented as cost-supply curves in GCAM. Specifically, production of biomass (and of all other crops represented in the model) depend on the respective crop yields and the amount of land allocated to crops. Regarding hydropower, we note that “hydropower availability” as mentioned in the manuscript refers to projected amounts of hydroelectricity generation (in EJ), which are set exogenously for all GCAM regions and time steps. Given that hydroelectricity generation in GCAM is a fixed input read in at the beginning of each simulation, there is no representation of distinct types and sizes of hydropower plants. It is also worth mentioning that hydropower production in the historical years (until 2010) is calibrated, noting that the International Energy Agency’s Energy Balances is the primary source for calibration of GCAM energy sector parameters. Lastly, we note that seasonality is not considered in any of the RE technologies represented in GCAM as noted in our response to reviewer’s comment #4.

We have revised the Methods section (“Climate impacts on renewables — model representation”) and Supplementary Note 3 to provide more details on (1) how climate impacts on the four renewables are modeled in GCAM; (2) how the climate-impact representations differs from the default (“no-climate impacts”) GCAM assumptions; and (3) how key model parameters are affected by the climate-impact model representations. In particular, we have clarified how hydropower is defined and modeled in GCAM, and the relationship between crop yield changes and biomass production. We have also decided to replace the term “hydropower availability” by “hydropower production” throughout the manuscript for the sake of clarity. Please see changes implemented in the Methods section (“Climate impacts on renewables — model representation”) and in the Supplementary Note 3. (Given the various edits and new text additions, these revised sections are currently very long to be shown here.)

16. Line 100 and 216: It is good that the authors have considered climate model uncertainty. I suggest that 3 GCMs is insufficient for a good representation of the climate model ensemble. For example, see comments in the ISI-MIP project about their choice of GCMs, and Carvajal et al 2017 <https://link.springer.com/article/10.1007/s10584-017-2055-4>. Please provide some justification for the choice of GCMs and validity of the method.

We agree with the reviewer that using more than 3 GCMs would be ideal to more adequately cover the large range of uncertainties across GCMs. We also acknowledge that the use of a larger ensemble of models represents a key opportunity for improvements in this research in future studies. The present research effort was limited by the availability of data on solar and wind supply-curves, which were produced by the “ISIPedia-energy protocol” project (referenced in Yalaw et al. 2020, reference 14 in the manuscript) based on the 4 GCMs from the ISIMIP2b dataset. This initial ensemble was further constrained by the crop yields dataset, based on data from the AGMIP project, which does not include yields data produced from the MIROC5 model (one of the 4 GCMs within the ISIMIP2b dataset). Although we would have preferred to use more GCMs, all three models (GFDL-ESM2M, HadGEM2-ES and IPSL-CM5A-LR) are state-of-the-art GCMs that have been widely employed by the climate change and climate-impact scientific community.

Further, the results of the uncertainty assessment do not seem to be considered in detail in the main text or SI. Comment is needed on what the uncertainty assessment shows regarding the robustness of the results and the resilience of the electricity system itself. For example, SI fig 11 shows that for some regions the difference in investments are positive from 1 GCM but negative from another. This uncertainty over the climate impacts would presumably have a significant implication for the extent to which system planners are able to account for climate impacts.

We have provided additional comments (new text included in the last paragraph of the “*Implications for power-sector capital investments*” section) to expand the discussion around the uncertainty of our results. New text reads:

“We recognize that the investment implications estimated in this analysis are inherently uncertain due to a wide range of outcomes from individual GCM-derived impacts (Supplementary Figs. 47–48). This wide range relates to the substantial uncertainties in GCM projections of variables such as precipitation, winds and shortwave solar radiation used to force the impact models employed herein. For this reason, uncertainties are high for all technology cases although the *NoCCS & NoNewNuc* exhibits, for most subregions, the greatest magnitudes of standard deviations associated with the more pronounced mean impacts in this scenario (Table 2). Overall, mean impacts estimated for Brazil, C. Am/Car., Mexico and S. Am. (N.) are associated with the largest spread of model outcomes (Supplementary Figs. 47–48). Although our ensemble of three climate runs is insufficient to cover the full range of uncertainties across GCMs, it provides initial estimates of overall bounds of economic impacts each region might experience. Importantly, we find larger confidence on investment projections for S. Am. (S), Argentina, Colombia and Uruguay, particularly under the *RCP26* cases, reflected in lower standard deviations (relative to their means) than in other subregions (Table 2) and agreement on the direction of the investment impact (Supplementary Figs. 47–48). Future research should employ a larger ensemble of models to improve overall confidence on the projected changes. Nonetheless, even employing considerably larger ensembles than the one used here, prior studies^{20,53} have highlighted the significant decision-making challenge arising from a large spread of individual model outcomes. To improve the resilience of energy systems in light of the large uncertainty in future climate projections, there are arguments supporting “uncertainty-management” methods⁵⁴ like adaptation strategies that are valid under alternative future

outcomes, diversify generation sources and consider a more decentralized small-scale energy structure⁵⁴⁻⁵⁷.”

17. Line 101: “Note that the RCP2.6 represents the lowest projected warming level among RCPs, consistent with the long-term goal of the Paris Agreement of keeping global warming likely below 2°C above pre-industrial temperatures¹². “

This statement is out of date as studies are now examining RCP1.9 and looking at scenarios where global warming is limited to 1.5C. Consider revising this sentence to say why you choose RCP2.6 and not higher or lower levels of warming.

The RCP2.6 was chosen because it is the lowest radiative forcing level available within the ISIMIP dataset (built upon GCM data forced by the RCPs considered within the IPCC AR5). Given our interest in studying climate impacts within the context of mitigation scenarios, the RCP2.6 is the lowest climate forcing level in which we have model inputs data available. We have revised the text, which now reads:

“Note that the RCP2.6 is the lowest projected warming level among the RCPs considered within the IPCC AR5 and ISIMIP, and is consistent with a global warming likely below 2°C above pre-industrial temperatures¹². The RCP2.6 allows climate impacts on renewables being studied in a context of strong climate change mitigation with substantial upscaling of renewable energy.”

18. Scenarios: the descriptions of the scenarios are clear and Table 1 is a useful summary. However, here and in the results below, consider centering the insights from scenario 3 (FullTech: Combined impacts), with description of the other scenarios alongside. It appears that the purpose of scenarios 1 and 2 (hydropower only) is to show that representation of climate impacts on hydro only is insufficient and gives very different results from when impacts on all RE techs are modelled. This therefore appears to be mainly a modelling contribution of the paper. In contrast, scenarios 3 and 4 (Combined impacts) provide insights on what could happen in the world – i.e. the climate impact of rcp2.6 and the (un-) availability of CCS and new nuclear. Consider re-focusing the paper on the technology scenarios with combined impacts, climate model uncertainty, and perhaps a scenario of stronger climate change for comparison (e.g. RCP4 or 6).

Thank you for the comments. On the last point, we have included new discussion on results produced under the RCP6.0 as suggested. We believe that these new results provide insights that improved the quality of the manuscript. With respect to the suggestions of refocusing the paper on model uncertainty and on the technology scenarios with combined impacts, we would like to mention that we have improved the discussion on uncertainty in response to reviewer’s comment #16, and provided additional information on the technology scenarios throughout the “Results” and “Discussion” sections (mostly in response to comments #19, 20, 22 and 35). However, we would prefer to keep the focus of the manuscript around the value of a comprehensive modeling of climate impacts on renewables.

Implications for electricity generation patterns

19. Line 115. Please add some description of the changes observed in scenario 1 or 3 relative to the ref scenario. I believe this is needed to help orientate the reader’s understanding of the

results.

Alternatively, restructure the results to centre the insights from scenario 3, with description of the other scenarios alongside, as described in comment 18.

In the first paragraph of the “Implications for electricity generation patterns” section, we have provided an overview of the mitigation scenarios explored in this study. We have opted for briefly describing both the *FullTech* and the *NoCCS & NoNewNuc* family of scenarios rather than focusing only on the *FullTech* scenarios. We have also included in the Supplementary Information, Supplementary Figures 7–10. These figures show differences in electricity generation mixes between the *RCP26* climate impact scenarios relative the GCAM *Baseline* scenario (although we are using the label “Baseline scenario”, the label “Reference scenario” is also largely used in many prior studies to refer to this same scenario). (Note that all scenario names changed relative to the first submission due to the inclusion of results pertaining to the RCP6.0 forcing). New text included in the first paragraph of the “Implications for electricity generation patterns” section reads:

“Consistent with prior literature on LAC decarbonization scenarios³⁶⁻⁴¹, our mitigation *RCP26* scenarios entail a significantly larger use of low-carbon energy sources and increased electrification of end-use sectors compared with a *Baseline* energy technology pathway (Supplementary Figs. 3-10). The *RCP26_FullTech* family of scenarios represents a diverse array of low-carbon technologies with bioenergy and natural gas plants equipped with CCS playing central roles in mitigation by supplanting the role of fossil-fuel based power generation, particularly, of natural gas, through 2100 (Supplementary Figs. 5, 7 and 9). Under the *RCP26_NoCCS & NoNewNuc* scenarios, emissions reductions in the power sector are achieved largely through the addition of solar and wind plants (Supplementary Figs. 6, 8 and 10). Uruguay stands out for a *Baseline* profile already predominantly reliant on RE, particularly on wind (Supplementary Fig. 3). In Uruguay, the mitigation scenarios lead to a replacement of bioenergy without CCS by bioenergy with CCS or wind depending on the technology pathway (Supplementary Figs. 7–10). As noted below, each energy technology pathway offers distinct technological alternatives for adaptation to climate impacts on RE.”

In any case, the structure of this section should be adjusted to clarify the impact of each scenario change i.e. to highlight the effects of modelling multiple climate impacts, the effects of climate change itself, the effects of the technology scenarios.

Please note our response to comment #18.

20. I suggest the results should describe key elements of the energy system technology transition, as well as the cumulative generation, as the timing of these changes is presumably very relevant for the investment requirements/costs.

We have included in the Supplementary Information, Supplementary Figures 15–22, which show projected electricity generation changes in the eight GCAM-LAC regions over distinct time frames from all climate-impact scenarios explored in this study. We have opted for keeping this material in the Supplementary Information because of the large amount of individual scenario outcomes given the current six climate-impact scenarios (including the two *Baseline* scenarios recently included to account for the RCP6.0) and eight GCAM-LAC regions to

consider. We have also included in the manuscript additional discussion around these new figures by focusing on the example of Argentina to highlight implications for decision-making. New text included reads:

“In all climate-impact scenarios, much of the differences in electricity generation tend to be more pronounced throughout the 2061-2100 period (Supplementary Figs. 29–36). Given the unique implications each subregion may face due to climate impacts on renewables, these results illustrate how distinct accounting of these impacts in IAMs may affect decision-making. For example, under the *Baseline* scenarios, Argentina is projected to experience a pattern of temporally increasing losses in hydroelectricity production (Supplementary Figs. 34 – left panels), which would require continuously improving adaptation plans. In this regards, modeling impacts only on hydropower implies that increased wind power generation would be among the portfolio of cost-effective adaptation options in Argentina. On the other hand, accounting for impacts in all renewables means that hydropower losses might be progressively exacerbated by losses in wind power generation, requiring a change in the course of power-sector adaptation plans in the country.”

21. Line 120: The terms ‘non-trivial’ and ‘negligible’ response are used throughout the paper. This should be defined if it implies a specific judgement on which responses are trivial and non-trivial.

In the revised manuscript version, the term “negligible” is only used once at the end of the second paragraph of the “Implications for electricity generation patterns” section, and in that context is simply a synonym for the word “small”. With respect to the term “nontrivial”, which is used twice throughout the manuscript, we recognize that there is a specific judgment in its use because the “nontrivial responses induced by the climate-impacted wind supply-curves” mentioned in the second paragraph of the “Implications for electricity generation patterns” section are, in fact, relative to the hydropower responses. We have clarified our usage of the term “nontrivial” in the first time it is used in the manuscript (second paragraph of the “Implications for electricity generation patterns” section). The revised sentence now reads:

“First, some LAC subregions (particularly Brazil, S. Am. N. and S. Am. S) show nontrivial responses induced by the climate-impacted wind supply-curves (that is, considerable changes in wind power production relative to the climate-impacted hydroelectricity generation; see Supplementary Figs. 11–14 and Supplementary Note 6 for details on how multiple interacting climate impacts combine and affect the modeled RE production).

22. Fig 1: The clarity of this figure is appreciated. Some information should be included to describe the starting technology mix of each country, or the mix in the ref scenario. As mentioned above, some more information on this is needed in the intro and SI.

As mentioned in our response to comment #11, we have included in the “Introduction” section information regarding the current technology mix in LAC and two additional figures in the Supplementary Information (Supplementary Figs. 2 and 3) that show the electricity generation pathways in all GCAM LAC regions under the RCP60_*Baseline* scenario. Moreover, in response to comment #19, we have provided an overview of the differences between the RCP26 climate impact scenarios explored in this study relative to the RCP60_*Baseline* scenario.

23. Line 146: “Note that the responses in hydroelectricity are consistent in all scenarios because the temporal evolution of hydroelectricity production per GCAM region is exogenously predetermined 25 (i.e., fixed for scenarios with and without impacts on hydropower).”

Clarification needed – I understand this sentence to mean that hydropower generation is not, and cannot be, impacted by climate change in the modelling. However, I see this can't be right, as the rest of the paper explains a climate change impact on hydro generation is modelled, and Fig 1 shows there is an impact on hydro generation in all the scenarios relative to the Ref scenario. Please adjust this to explain why the climate impact on hydropower is the same in all scenarios relative to the Ref. i.e. please clarify the meaning of “the temporal evolution of hydroelectricity production is exogenously predetermined.” Fixed to what levels and why?

We thank the reviewer for calling attention for the lack of clarity in this aspect. There are two main points to note: (1) how hydropower is modeled in GCAM under default no-climate impacts assumptions, and (2) how default assumptions are modified when climate impacts on hydropower are incorporated into the model.

With respect to the first point, we note that the meaning of “the temporal evolution of hydroelectricity production is exogenously predetermined” is that the quantity of hydroelectricity produced in all future years (in EJ) is set as an external (fixed) input into the model. These default projections of hydroelectricity production vary by GCAM region and are based on the economic and technical potentials estimated by the International Hydropower Association.

To account for climate impacts on hydropower, the above-mentioned default assumptions of hydroelectricity production are modified based on trends derived from the global hydrological model Xanthos forced by GCMs data (further details in Supplementary Note 3) with the goal of capturing long-term mean climate effects. As a result, each climate forcing level (RCP26 and RCP60) is characterized by specific hydropower input assumptions. However, these quantities differ from those in scenarios without climate impacts on hydropower (which rely on the GCAM default assumptions of hydroelectricity production). We have revised this sentence as well as Methods (see our response to reviewer's comment #15) and Supplementary Note 3 to provide the needed clarification. Revised sentence now reads:

“Note that regional responses in hydropower are consistent in scenarios with the same climate forcing because the temporal evolution of hydropower in GCAM follows predetermined input assumptions on how much hydroelectricity each region will produce per time step. Each climate forcing level (RCP2.6 and RCP6.0) has distinct assumptions (see Methods and Supplementary Note 3 for details on the modelling of hydropower and climate impacts in GCAM).”

Implications for power-sector capital investments

24. Line 163: On average, total capital investment needs in LAC increase by approximately USD 17–114 billion compared to the No-Climate impacts scenarios (Table 2). State here what these costs represent (cumulative investments over the period 2020 -2100?).

Yes, costs represent cumulative investments over the period 2020-2100. Revised sentence reads:

“On average, cumulative total capital investment needs in LAC over the 2020–2100 period increase by approximately USD 12–114 billion compared to the *No-Climate impacts* scenarios (Table 2).”

25. Line 168. This is an interesting finding, and the context and comments in this paragraph are very useful. The investments do seem small compared to the recent historic investments in the region. I recommend moving this comparison to the discussion and commenting on whether this indicates climate change impacts should be accounted for by policy makers/system planners, or if they are of little concern. Consider the uncertainty on the results alongside this.

Thank you for the comment and suggestion. Although the suggestion is interesting, we would prefer to keep the text in the original position (same for the uncertainty discussion) as we believe it fits very well where it currently is. Regarding the requested comment on whether our relatively low estimates indicate that climate change impacts should be accounted for by decision-makers, we believe that the sentence immediately following the one highlighted by the reviewer provides our assessment on the importance of accounting for these impacts. Specifically we are referring to the sentence below:

“Although these additional investments seem small, they could imply significant challenges for the developing economies in LAC, where resources for public investments are scarcer, and private financing costs (closely linked to perceptions of the quality of institutions and associated investment risks^{33,52}) are generally higher compared to the developed world.”

26. Line 185: “The regional patterns in additional investments largely follow the effects induced by the climate-impacted RE inputs on electricity production outlined in Fig. 1” This sentence is a bit unclear. Consider rephrasing it to something like “The country-level additional investment results largely reflect the changes to the electricity technology mix shown in Fig 1.”

Furthermore, is the headline message that where climate change increases the primary resource, the model indicates that the least cost option is to install more of that technology so investments are directed towards that technology, and vice versa? If so, I recommend adding a sentence to this effect.

We thank the reviewer for calling attention for the lack of clarity of the sentence. In this sentence, we have only intended to highlight that the changes in capital investments shown in Fig. 3 are largely in line with the changes in electricity generation shown in Fig. 1. Regarding reviewer’s comment on “where climate change increases the primary resource, the model indicates that the least cost option is to install more of that technology so investments are directed towards that technology, and vice versa”, this tends to be true, but it does not necessarily occur in all circumstances. The reason is the amount of interactions throughout the model, which affect the economic competitiveness of generating technologies relative to others. The latter is important because, in GCAM, energy-technology decisions are made based on technology costs (we have included some new information on this decision-making approach in the “Methods” Section, “The Global Change Analysis Model (GCAM)” subsection in response to comment #41, and on the “Climate impacts on renewables — model representation” subsection). We have rephrased the sentence as suggested, which now reads:

“The regional differences in investments largely reflect the changes to the electricity technology mix shown in Fig. 1”

27. Line 191: Ensure all region names are consistent throughout the paper. E.g. C. Am. C. or C. Am/Car

This has been revised throughout the entire manuscript.

28. Line 210: “the negligible role of climate impacts” Clarify if this means the climate impacts on non-hydro renewables are small, or if non-hydro renewables account for a small portion of the electricity generation mix

The sentence has been revised to:

“Exceptions are Argentina, where reductions in total capital investments in the *RCP26 Combined impacts* scenarios are considerably larger than in the *Hydropower* scenarios due to lower wind capacity requirements, and Colombia and Uruguay, where total investment requirements are consistent in both *RCP26* climate-impact scenarios because climate impacts on non-hydropower renewables do not play important roles (recall Fig. 1).”

29. Table 2 Caption: Clarify if the total capital investments include all capital investments for the energy system, or just the electricity sector, or just RE technologies, or just utility scale RE etc.

Table 2 includes total capital investments specifically in the electric power sector. Table 2 caption has been revised to:

“**Table 2.** Regionally aggregated changes in total capital investments in the LAC electric power sector under the Combined impacts scenarios. Changes represent the mean value (absolute and percentage) across GCMs (the standard deviation of the absolute model mean change is also shown), and are calculated using cumulative investments in the 2020 – 2100 period. Changes are relative to the No-Climate impacts scenarios (i.e., positive values mean that scenarios with climate impacts on renewables show increased costs). The corresponding results for the period 2020 – 2050 are provided in Supplementary Table 10.”

Discussion

30. Line 248: I understand this sentence to mean that more wind capacity (incentivized by the positive impact of climate change on the wind resource) implies higher investments in wind, which I think is self-evident. Presumably the higher wind capacity displaces one or more other technologies. So here, the total capital investments should be compared rather than the investments in just wind.

We have revised the text to improve clarity. The goal here is to highlight the potential opportunities created by enhanced wind power production, and the need of specific policies to overcome the financial barriers created by the high upfront capital costs of wind technologies (and of renewables in general). Revised text reads:

“This emerges as a strategic opportunity for decarbonization and diversification of regional power mixes. However, the high upfront capital expenditures of wind technologies (and of renewables in general) represent a critical financial barrier to RE deployment, particularly in developing economies, requiring specific policies to create favorable financing conditions^{4,33}.”

31. Line 250: “In this context, Brazil (which has led wind energy expansion in LAC) and Chile (part of S. Am. (S)) are currently more advanced in creating favorable institutional frameworks for investments in renewables than other countries within S. Am. (S) and S. Am. (N)²⁸.” It is unclear how this is related to the previous statement. Discussion of how the results of these study compare to historic rates of capacity expansion, and the relevance of institutional frameworks would be extremely valuable and require more space devoted to them.

Thank you for pointing out the lack of clarity of this sentence. As mentioned in our response to comment #30, we have revised text to improve clarity. We also thank the reviewer for the suggestion on expanding discussion on the relevance of institutional frameworks. Despite clearly relevant within the context of power-sector investments in the developing economies of LAC, a deeper discussion on the role of institutional frameworks lies beyond the main purpose of this article. With respect to the suggestion on comparing results of our study to historic rates of capacity expansion, we have prepared such a comparison using data from Energy Statistics Database of the United Nations Statistics Division (available at <http://data.un.org/Explorer.aspx>). This comparison was included in the Supplementary Information (Supplementary Figures 23–30) given that we are limited in space to add this material in the main text. The inclusion of these new figures is indicated in the main text, first paragraph of the of the “Implications for power-sector capital investments” section, as:

“Power-sector capital investments depend on how much generating capacity is installed or retired over time per technology and the marginal costs of building capacity from each technology (Methods and Supplementary Note 5). Hence, the climate-induced alterations in electricity production patterns discussed so far would have implications for regional capital investment needs through changes in generating capacity (Supplementary Figs. 37–44 compare how our future estimates of generating capacity compare with historical rates).”

32. Line 256: “diversified mix of generating technologies with sizable contributions from CCS and, thus, lower exposure to climate impacts on renewables,” Check the rigour of this sentence. Do you mean CCS on fossil fueled thermal power stations? If so, it is the continued use of fossil fuels, facilitated by CCS, which reduces the exposure of the system to climate impacts on renewables.

To conclude that a diversified mix of renewables reduces the exposure to climate impacts, you would need to examine the correlation of climate impacts between different RE technologies in each country, analysis of which is not prominent in this paper.

Thank you for calling attention to the lack of clarity of the sentence. Yes, we meant to say “CCS applied to thermal power generation” as a mean to contribute toward a more diversified mitigation pathway (illustrated by the *RCP26_FullTech* scenario), which reduces the exposure of the system to climate impacts on renewables compared to a pathway more heavily reliant on renewables such as the *RCP26_NoCCS & NoNewNuc* case. We have revised the sentence to improve clarity, which now reads:

“On the one hand, a mitigation pathway based on a diversified mix of generating technologies with sizable contributions from fossil-fueled plants with CCS, as illustrated by the *RCP26_FullTech* scenario, reduces the exposure of the power system to climate impacts on renewables, and may alleviate (or avoid) the necessity of raising investments.”

33. Line 260: “decarbonizing LAC’s power sector largely through climate-sensitive solar and wind technologies may increase risks of higher capital investment requirements. “ This should be tied back to the results of this study, e.g. ‘as was demonstrated in the comparison of X and Y scenarios for A/b/c countries.”

We have revised the sentence, which now reads:

“On the other hand, decarbonizing LAC’s power sector largely through climate-sensitive solar and wind technologies may increase risks of higher capital investment requirements, as shown in Table 2 for most LAC regions under the *RCP26_NoCCS & NoNewNuc: Combined impacts* scenario.

34. Line 262: “One key point is the significantly lower capacity factors of intermittent renewables compared with other technologies, which means that both RE sources require more capacity to generate the same amount of electricity than other technologies, such as fossil fuels with CCS. “ This is technically true but not highly relevant, as capacity is not fully equivalent for different technologies – it is more interesting to discuss total system investment costs or the use of land or natural resources, or efficiency of the system.

We have revised the sentence, which intends to clarify the reason results under the *NoCCS & NoNewNuc: Combined impacts* scenario (now called *RCP26_NoCCS & NoNewNuc: Combined impacts*) led to higher needs of capital investment in most regions. We believe this information helps readers to understand article’s results. Revised text reads:

“These larger increases relate to the lower capacity factors of intermittent renewables compared with fossil fuels with CCS technologies deployed in *RCP26_FullTech: Combined impacts* scenario. This means that intermittent renewables require more generating capacity per unit of electricity produced compared with fossil-fuel technologies with CCS (Supplementary Note 5 shows how capacity factors are used to compute capital investments in our methodology).”

35. There is little description of the impact of the two technology scenarios. These are a very interesting contribution of this paper so should be highlighted and discussed in more detail.

Thank you for the comment. We have clarified the role of the energy technology scenarios by including additional discussion at the end of the first paragraph of the “Discussion” section. New text included reads:

“GCAM results highlight regionally differentiated impacts across LAC power grids due to a combination of vulnerabilities specific to each generation mix and large spatial variability of climate change impacts across LAC. We explore the first component through distinct technology pathways, showing that the generation portfolio plays an important role in alleviating or exacerbating increasing pressure on capital investments due to climate-attributable effects on

renewables. Since each energy technology pathway affects the availability of technology replacement options (each of them characterized by specific costs of installing generating capacity), implications for total capital investments differ markedly.”

36. The interplay between elements of the electricity system should be explored further. i.e. which technologies displace others in each of the scenarios, and how do the climate impacts combine?

Thank you for the suggestion. We agree that the point raised by the reviewer is valid. However, we would like to mention that the main goal of this article is to emphasize the consequences for the modeling of electricity generation and capital investment requirements due to the two different representations of climate impacts on renewables rather than to focus on the technology pathways by themselves. As mentioned in the revised manuscript, the energy technology pathways serve to analyze climate impacts on renewables in the context of very distinct technological replacement options for adaptation. Moreover, at this point, we are constrained by the limit of words allowed by the journal (5000), making it difficult to further detail the technology interplays among each of the scenarios. However, we believe that due to the recent additions to the manuscript suggested by the reviewer in comments #18, 19, 20, 22 and 35, readers can now have a better understanding on the role of the energy technology scenarios and technology interplays associated with each of them. We also believe that Supplementary Figs. 11–14 and Supplementary Note 6 provide useful information for the understanding on how climate impacts combine within our modeling framework and how renewable energy production is affected in response to the multiple-impacts approach. In the second paragraph of the “Implications for electricity generation patterns” section, we highlight the role of the supplementary material mentioned above. New text included reads:

“; see Supplementary Figs. 11–14 and Supplementary Note 6 for details on how multiple interacting climate impacts combine and affect the modeled RE production).”

37. The authors constructed the scenarios and modelling to examine the impact on the energy system if each of the climate impacts is modelled separately then in combination. This is very interesting but is not discussed in much detail. What can be deduced from this, in terms of the vulnerability of electricity systems which are more or less diversified in a changing climate?

We have revised discussion on the implications of our results in terms of vulnerabilities of regional generation portfolios. Specifically, we discuss the fact that a diversified generation portfolio emerges in our results as less vulnerable to climate impacts on renewables albeit requiring careful consideration of potential detrimental effects on mitigation goals. This revision was made on the fifth paragraph, which now reads:

“The growing trends in LAC’s power-sector capital investment requirements reported under multiple RE impacts and technology configurations suggest challenges for the planning of low-carbon capacity additions. On the one hand, a mitigation pathway based on a diversified mix of generating technologies with sizable contributions from fossil-fueled plants with CCS, as illustrated by the *RCP26_FullTech* scenario, reduces the exposure of the power system to climate impacts on renewables, and may alleviate (or avoid) the necessity of raising investments. However, CCS technologies are not mature, nor have they been widely deployed commercially

yet. On the other hand, decarbonizing LAC's power sector largely through climate-sensitive solar and wind technologies may increase risks of higher capital investment requirements, as shown in Table 2 for most LAC regions under the *RCP26_NoCCS & NoNewNuc: Combined impacts* scenario. These larger increases relate to the lower capacity factors of intermittent renewables compared with fossil fuels with CCS technologies deployed in *RCP26_FullTech: Combined impacts* scenario. This means that intermittent renewables require more generating capacity per unit of electricity produced compared with fossil-fuel technologies with CCS (Supplementary Note 5 shows how capacity factors are used to compute capital investments in our methodology). Although the value of diversifying the energy portfolio has been recognized as a mean to achieve climate resilient power systems⁵⁵, it is crucial that energy planners identify strategies that do not jeopardize climate goals. In this regards, a mixture of renewable and non-renewable energy sources, albeit less vulnerable to climate impacts on renewables, can dampen mitigation efforts unless CCS technologies become technically viable and cost-competitive and/or comprehensive emissions reductions actions are implemented. Regarding the latter, one alternative might be to focus more heavily on reducing emissions from land and agricultural systems and on enhancing terrestrial sinks for carbon in future decades. This is particularly relevant in LAC where land-related GHG emissions make up a significant share of total emissions⁵⁸.”

38. Finally, the implications for the electricity systems in these countries should be discussed. What are the key messages for the system planners? Can the results be discussed in terms of whether they indicate certain vulnerabilities, or opportunities to increase resilience? And the cost of mitigating the risks?

We highlight below instances taken from the revised “Discussion” section where we have discussed the overall key insights of practical implications for system planners, system vulnerabilities, opportunities to increase resilience, and the cost of mitigating the risks.

Second paragraph (highlights risks of misrepresentation of climate change effects on the electric power sector if climate impacts on all renewables are not accounted for with potential detrimental consequences to the vulnerability of regional energy systems):

“The key overarching insight from all scenarios explored herein is the risk of misrepresentation of climate change effects on the electric power sector if climate impacts on all renewables are not accounted for. This is particularly evident for the energy pathway with the most pronounced intermittent renewables deployment (i.e., the *NoCCS & NoNewNuc*), characterized by greatly underestimated capital investment requirements across most of the LAC region when climate impacts only on hydropower are considered. Such an underestimation may result in enhanced power-sector vulnerabilities to climate change.”

Fourth paragraph (highlights opportunities to increase resilience of regional electricity system – through diversification of energy sources – that may arise under favorable climate impacts on wind):

“Our results also highlighted an overlooked angle related to the fact that climate impacts on wind at the 2°C warming level can positively affect power production in certain LAC subregions

(Brazil, S. Am. (N) and S. Am. (S)). This emerges as a strategic opportunity for decarbonization and diversification of regional power mixes.”

Fifth paragraph (highlights the larger vulnerability of energy technology pathways with more pronounced deployment of renewables versus diversified technology pathways; emphasizes that the cost of mitigating risks of increased capital investment requirements through diversified energy portfolios may be a failure in achieving climate goals unless specific actions to address such a risk are taken).

“The growing trends in LAC’s power-sector capital investment requirements reported under multiple RE impacts and technology configurations suggest challenges for the planning of low-carbon capacity additions. On the one hand, a mitigation pathway based on a diversified mix of generating technologies with sizable contributions from fossil-fueled plants with CCS, as illustrated by the *RCP26_FullTech* scenario, reduces the exposure of the power system to climate impacts on renewables, and may alleviate (or avoid) the necessity of raising investments. However, CCS technologies are not mature, nor have they been widely deployed commercially yet. On the other hand, decarbonizing LAC’s power sector largely through climate-sensitive solar and wind technologies may increase risks of higher capital investment requirements, as shown in Table 2 for most LAC regions under the *RCP26_NoCCS & NoNewNuc: Combined impacts* scenario. These larger increases relate to the lower capacity factors of intermittent renewables compared with fossil fuels with CCS technologies deployed in *RCP26_FullTech: Combined impacts* scenario. This means that intermittent renewables require more generating capacity per unit of electricity produced compared with fossil-fuel technologies with CCS (Supplementary Note 5 shows how capacity factors are used to compute capital investments in our methodology). Although the value of diversifying the energy portfolio has been recognized as a mean to achieve climate resilient power systems⁵⁵, it is crucial that energy planners identify strategies that do not jeopardize climate goals. In this regards, a mixture of renewable and non-renewable energy sources, albeit less vulnerable to climate impacts on renewables, can dampen mitigation efforts unless CCS technologies become technically viable and cost-competitive and/or comprehensive emissions reductions actions are implemented. Regarding the latter, one alternative might be to focus more heavily on reducing emissions from land and agricultural systems and on enhancing terrestrial sinks for carbon in future decades. This is particularly relevant in LAC where land-related GHG emissions make up a significant share of total emissions⁵⁸.”

39. Line 262 – 276. These are interesting but quite general statements about renewables and energy systems, without direct relevance to the scope of this study. These should be removed to make space for more in depth discussion on the technology scenarios, interplay between elements of the electricity system and wider implications of the investments results.

As noted in our response to reviewer’s comments #37 and 38, we have revised these sentences with the main goal of emphasizing practical implications of our results for decision-making in the context of LAC. Revised text reads:

“These larger increases relate to the lower capacity factors of intermittent renewables compared with fossil fuels with CCS technologies deployed in *RCP26_FullTech: Combined impacts* scenario. This means that intermittent renewables require more generating capacity per unit of

electricity produced compared with fossil-fuel technologies with CCS (Supplementary Note 5 shows how capacity factors are used to compute capital investments in our methodology). Although the value of diversifying the energy portfolio has been recognized as a mean to achieve climate resilient power systems⁵⁵, it is crucial that energy planners identify strategies that do not jeopardize climate goals. In this regards, a mixture of renewable and non-renewable energy sources, albeit less vulnerable to climate impacts on renewables, can dampen mitigation efforts unless CCS technologies become technically viable and cost-competitive and/or comprehensive emissions reductions actions are implemented. Regarding the latter, one alternative might be to focus more heavily on reducing emissions from land and agricultural systems and on enhancing terrestrial sinks for carbon in future decades. This is particularly relevant in LAC where land-related GHG emissions make up a significant share of total emissions⁵⁸.”

Methods

40. Line 474: I suggest this should say ‘socio-economic drivers’ or parameters or trends

This has been change to “socio-economic drivers” as suggested.

41. Line 477: Please clarify this sentence. In what ways does GCAM find the ‘most technically feasible combination of technologies...’? Is it in fact the least cost combination, subject to exogenous constraints which represent the technically feasible costs and availabilities of primary energy resources and supply and demand side technologies?

Yes, but it is important to point out that GCAM decision making relies on a logit formulation that prevents the least-cost competing options from gaining entire shares of the modeled markets. More specifically, within limits imposed by its inputs (costs, current and future technologies, efficiencies, availability of resources, etc.), GCAM iteratively searches for the set prices that equilibrates supplies and demands in all sectors. This process aims at finding a solution that minimizes costs or maximizes profits (as in the case of the land sector). However, given the logit formulation implemented in GCAM, preference among competing options depend on their costs or expected profit rates (Calvin *et al* 2019). This means that the least-cost or most profitable options capture the largest shares of the markets, but the other options also gain some market share. We have complemented the text with additional information to provide the requested clarification. Revised text now reads:

“This allows a multi-sectoral assessment of implications in a way that the model solution represents the least-cost and most technically feasible combination of existing technologies and resources per region. More specifically, given limits imposed by its inputs (costs, current and future technologies, efficiencies, availability of resources, etc.), GCAM iteratively searches for the set of prices that equilibrates supplies and demands in all sectors. This process aims at finding a solution that minimizes costs or maximizes profits (as in the case of the land sector). However, decision-making in GCAM relies on a logit-choice formulation (Eq. 2), in which preference among competing options depend on their costs (see Eq. 1) or expected profit rates³⁰. Although the least-cost or most profitable options capture the largest shares of markets, the other options also gain some market share as explained in the following subsection. Further details on the GCAM are provided in the Supplementary Note 2.”

Reference:

- Calvin, K. et al. GCAM v5. 1: representing the linkages between energy, water, land, climate, and economic systems. *Geoscientific Model Development* 12, 677-698 (2019).

42. Line 482: I suggest this should say ‘socio-economic drivers’ or parameters or trends

This has been change to “socio-economic drivers” as suggested.

43. Line 482: I suggest this should say “the disaggregation of Uruguay” because “break-up” sounds like Uruguay itself has been split up.

This has been revised as suggested.

44. Line 482: For the sake of being re-producible, I suggest some of these improved parameters should be summarized in the SI.

We have included a new table in the Supplementary Information (Supplementary Table 5), which contains a list of updates in the GCAM-LAC model.

45. Line 492: Specify which part of the SI

We are referring to Supplementary Note 3. This has been revised.

Supplementary Information

46. SI Figure 1 - I am unclear on the meaning of these two charts. Please clarify the legend. What does ‘share of RE ...relative to the rest of the world’ mean? I think the share of RE in a region’s power generation system should be in terms of GWh of RE generation/GWh total generation...?

The reviewer is correct in interpreting “share of RE in a region’s power generation system” in terms of GWh of RE generation/GWh of total generation. In the top chart, we meant to express the percentage of renewable electricity generation in total electricity generation in LAC as a whole (~55%) compared to the same percentage computed for the average of all other regions outside LAC (~26%). The legend has been revised and now reads:

“Supplementary Figure 1: Share of renewable energy (bioenergy, geothermal, hydropower, solar and wind) in total electricity generation: in the Latin American and the Caribbean (LAC) region compared to the average of the rest of the world (top); and by individual regions (bottom). The share of renewables in power generation was computed as total renewable electricity generation divided by total electricity generation expressed in relative (%) terms. [Notes: (1) 33 geopolitical regions are represented in the GCAM-LAC model version with 8 regions in LAC: Argentina, Brazil, Central America and Caribbean, Colombia, Mexico, South America Northern, South America Southern, and Uruguay; (2) Source: GCAM-LAC total electricity generation by region in 2010 (last calibrated year¹).]”

Responses to Reviewer 2:

Reviewer #2 (Remarks to the Author):

This is a very interesting paper with a lot of ambition and could provide some good insights into the effects of climate change on renewable energy resources, and could potentially be useful for long-term energy planning exercises. The work explored in the paper could also stimulate follow-up work in more micro (national) studies. There are some minor areas for the authors to carry out their adjustments and/or provide their responses:

Thank you very much for the positive comments and very constructive suggestions. We have detailed below the revisions and additions to the manuscript in response to them.

1. The authors discuss the existence of literature around the climate change impacts on hydropower; but there is little about the climate change impacts for solar and wind resources. The question then is how do we deal with the high level of uncertainty associated with solar and wind? A bit more discussion is needed here.

There is some literature discussing adaptation strategies in light of the unavoidable large GCM uncertainties given that such an issue is not expected to be significantly improved over the short-term. This literature has pointed out that rather than expecting scientific improvements in climate modeling to considerably reduce the high level of uncertainties in climate projections, decision-making should consider adaptation approaches that incorporate these uncertainties. This type of approach was referred to as “uncertainty-management” methods by Hallegatte (2009). In the manuscript, we have provided additional comments (new text included in the last paragraph of the “*Implications for power-sector capital investments*” section) to expand this discussion around the uncertainty of our results. New text reads:

“We recognize that the investment implications estimated in this analysis are inherently uncertain due to a wide range of outcomes from individual GCM-derived impacts (Supplementary Figs. 47–48). This wide range relates to the substantial uncertainties in GCM projections of variables such as precipitation, winds and shortwave solar radiation used to force the impact models employed herein. For this reason, uncertainties are high for all technology cases although the *NoCCS & NoNewNuc* exhibits, for most subregions, the greatest magnitudes of standard deviations associated with the more pronounced mean impacts in this scenario (Table 2). Overall, mean impacts estimated for Brazil, C. Am/Car., Mexico and S. Am. (N.) are associated with the largest spread of model outcomes (Supplementary Figs. 47–48). Although our ensemble of three climate runs is insufficient to cover the full range of uncertainties across GCMs, it provides initial estimates of overall bounds of economic impacts each region might experience. Importantly, we find larger confidence on investment projections for S. Am. (S), Argentina, Colombia and Uruguay, particularly under the *RCP26* cases, reflected in lower standard deviations (relative to their means) than in other subregions (Table 2) and agreement on the direction of the investment impact (Supplementary Figs. 47–48). Future research should employ a larger ensemble of models to improve overall confidence on the projected changes. Nonetheless, even employing considerably larger ensembles than the one used here, prior studies^{20,53} have highlighted the significant decision-making challenge arising from a large spread of individual model outcomes. To improve the resilience of energy systems in light of the

large uncertainty in future climate projections, there are arguments supporting “uncertainty-management” methods⁵⁴ like adaptation strategies that are valid under alternative future outcomes, diversify generation sources and consider a more decentralized small-scale energy structure⁵⁴⁻⁵⁷.”

Reference:

Hallegatte, S. Strategies to adapt to an uncertain climate change. *Global Environmental Change* **19**, 240-247, doi:<https://doi.org/10.1016/j.gloenvcha.2008.12.003> (2009).

2. Why only use the RCP 2.6? It may be that you would like to be consistent with the Paris Agreement, but what happens in the possible chance of higher temperatures. Presumably this would mean different level of impact on renewable resources. Why did you lock yourselves to the stringent RCP 2.6? It would have been very valuable to also look at a less stringent pathway (RCP), which is not only possible, but also may offer additional analysis to decision makers that delayed action on climate change may lead to heavier renewable energy costs in the future.

Thank you very much for the suggestion. Indeed, the original goal was to look into power sector implications from climate change impacts on renewables within the context of mitigation scenarios. However, we agree with Reviewer 2 that adding a higher climate forcing level to the analysis brings more value to the study. Hence, we have included in the results section additional discussion around a scenario forced by climate-impact inputs produced under the RCP6.0. Please see below new text added to the manuscript.

New text added to the “Implications for electricity generation patterns” section:

“Fig. 1 also emphasizes implications from distinct warming levels. A salient response from the *Hydropower* scenarios (columns 1, 2 and 3 in Fig. 1) is an overall deterioration of hydroelectricity production under the RCP6.0. All regions, except for Colombia, experience enhanced reductions in cumulative generation, shifts from generation gains toward losses or less pronounced positive impacts compared to the *RCP26 Hydropower* scenarios. C. Am/Car., Mexico, S. Am. (N) and Argentina emerge as particularly prone to negative impacts on hydropower as the severity of climate change increases. In these regions, a potential adaptation strategy assessed by GCAM might be to increase fossil fuel based generation (particularly natural gas), which can exacerbate the initial climate change signal via increments in fossil fuel emissions. A comparison between the *RCP60_Baseline: Hydropower* and *RCP60_Baseline: Combined impacts* scenarios (columns 1 and 4 in Fig. 1) reinforces the importance of detailed considerations of multiple impacts, which is particularly prominent in C. Am/Car., Mexico and Argentina. Again, the combination of impacts on hydropower and wind are the leading drivers of the compounding effects on electricity generation, however the direct effects on electricity generation changes induced by the RCP6.0 wind supply curves tend to be less pronounced than those induced by the RCP2.6 curves (Supplementary Figs. 11–14). This is particularly true for Brazil, S. Am. (N) and S. Am. (S). As a result, these regions experience less pronounced gains in wind-based generation under the *RCP60_Baseline: Combined impacts* relative to the *RCP26_FullTech: Combined impacts* case. It is important to note that these distinct outcomes must not be entirely attributed to the climate change signal due to the role of the energy technology pathway by itself. Specifically, under the *RCP60_Baseline* scenario, the effects

produced by the wind supply curves (shown in Supplementary Figs. 17–19) on wind power generation originate from the lower ends of the curves as wind power needs are not so prominent in this scenario. Conversely, energy-technology pathways like the *FullTech* and, in particular, the *NoCCS & NoNewNuc* rely considerably more on wind power to fulfill climate goals, thus suffering stronger influence from upper portions of the supply curves, in which differences among climate-impacted curves are more pronounced.”

New text added to the “Implications for power-sector capital investments” section:

“Under the *RCP60_Baseline* scenarios, there are also examples in which the *Hydropower* case do not show lower investment requirements relative to the *Combined impacts case* – Mexico and Argentina. However, investment estimates in these subregions under the distinct climate-impact modeling approaches differ markedly.

Although it could be expected that the *RCP60_Baseline: Combined impacts* scenario would yield considerably larger needs of capital investments in face of more severe climate impacts, we find that investment changes under the *RCP60_Baseline: Combined impacts* scenario are predominantly lower than or close to those in the *RCP26_FullTech: Combined impacts* case (Fig. 3 and Table 2). One key aspect is the overall low reliance of the *Baseline* pathways on RE as pointed out earlier. Under the *RCP60_Baseline* scenarios, no cost penalties are imposed for emitting fossil fuels, meaning that it is economically attractive to compensate part of renewable-based generation losses by fossil fuels without CCS, typically less capital-intensive than low-carbon options. This dynamic is more evident in Argentina and Mexico. These results then emphasize the role of the energy technology strategy in shaping the overall power-sector vulnerability to climate impacts on RE.”

3. How were the decisions for the scenarios made? and how were the calculations for the estimates done? And what data were used?

Thank you for calling attention for the need of clarification on the methodology and data used. We have conducted major revisions in various parts of the manuscript to provide more details and clarifications on the methods and data employed. Please see changes implemented in the Experimental design section, Methods section (“Climate impacts on renewables – model representation”) and in the Supplementary Note 3. (Given the various edits and new text additions, these revised sections are currently very long to be shown here.)

4. At the end of the discussion, the authors make the point that 'finer resolution, multi-impact integrated framework' studies would be needed to support national level decision makers. It would be useful to say a bit more on this to provide some guidance on what such national studies can gain from your study and how they can build on it.

Thank you for the suggestion. We have incorporated additional discussion based on a recent example of finer-resolution integrated assessment study (Khan *et al* 2020), which included scenarios with climate impact representations for hydropower and crop yields. Although the study certainly represents an important advance, we argue that future higher-resolution integrated analyses would need to incorporate a more comprehensive framework of climate change impacts as the one presented in our study to avoid the risk of misrepresentation of climate change effects on the energy sector. We can also envision other regional integrated assessment tools such as the

Platform for Regional Integrated Modeling and Analysis (PRIMA; described in Kraucunas *et al* 2014) and the Nexus Solutions Tool (NEST; described in Vinca *et al* 2020) being benefited from the multi-impacts approach perspective highlighted in our study.

New text included in the manuscript (“Discussion” section) reads:

“Hence, further research is needed to develop a finer-resolution multi-impact integrated framework that supports decision-making at sub-national scales. For example, Zarrar *et al*⁴⁴ contribute to fill such a gap by coupling GCAM and a suite of modeling tools to downscale GCAM projections (part of them including climate impacts on hydropower and agricultural crop yields) onto a grid. This framework was used for a multi-sector assessment of planned policies in Uruguay at a sub-basin scale. Given the possibility of misrepresentation of climate change effects on the power sector highlighted in our results, future high-resolution integrated assessments can benefit from a more comprehensive representation of climate change impacts like the one introduced in this study.”

References:

- Zarrar, K. et al. Integrated energy-water-land nexus planning to guide national policy: an example from Uruguay. *Environmental Research Letters* (2020).
- Kraucunas, I., L. Clarke, J. Dirks, J. Hathaway, M.I. Hejazi, K. Hibbard, M. Huang, C. Jin, M. Kintner-Meyer, K. Kleese van Dam, R. Leung, H. Li, R. Moss, M. Peterson, J. Rice, M. Scott, A. Thomson, N. Voisin, and T. West (2014). Investigating the Nexus of Climate, Energy, Water, and Land at Decision-relevant Scales: The Platform for Regional Integrated Modeling and Analysis (PRIMA). *Climatic Change*, <https://doi.org/10.1007/s10584-014-1064-9>.
- Vinca, A., Parkinson, S., Byers, E., Burek, P., Khan, Z., Krey, V., Diuana, F. A., Wang, Y., Ilyas, A., Köberle, A. C., Staffell, I., Pfenninger, S., Muhammad, A., Rowe, A., Schaeffer, R., Rao, N. D., Wada, Y., Djilali, N., and Riahi, K.: The NExus Solutions Tool (NEST) v1.0: an open platform for optimizing multi-scale energy–water–land system transformations, *Geosci. Model Dev.*, 13, 1095–1121, <https://doi.org/10.5194/gmd-13-1095-2020>, 2020.

Responses to Reviewer 3:

Reviewer #3 (Remarks to the Author):

Review on

‘Power sector investment implications of climate impacts on renewable resources in Latin America and the Caribbean’

The paper assesses the impact of climate induced changes to renewable energies on the overall energy system capacity.

Overall, I consider the paper clearly structured and well written and the results add to the overall energy picture on climate change impacts.

Thank you for the encouraging feedback. Please see our responses to your comments below.

Being more familiar with electricity modeling, I nevertheless have one fundamental concern with the approach and resulting interpretation. If I understand the model approach correctly the generation mix is based on costs structures accounting for the fact that renewables have a lower overall output due to intermittency. However, no detailed time structure on demand or supply sides is included (i.e. the capacity is defined by the total output and the capacity factor of each technology).

While such an approach is well suited for dispatchable technologies it does not really account for intermittent wind and solar generation. As those are central for future electricity systems their specifications need to be accounted when making future assessments.

The reviewer's general understanding is correct. However, as we clarify further, there are several nuances to the modeling of power generation and renewable intermittency in particular. We discuss those in detail in our responses below.

The main problem arising from an average (per year) approach is the simple fact that weather fluctuations (= the intermittency of wind and solar) as well as fluctuations between years (windy vs. less wind years) will require a different capacity approach to keep the electricity system stable and working.

Let's make an example: if there would be an unlimited zero cost solar PV option in your model, that should be the first and only technology to be installed (up to the point where total demand = total generation of this PV technology) if I understand the model correctly. If this is the case than its obvious that the model has a shortcoming, as such a system would not provide sufficient electricity at all times.

This general problem will lead to the requirement to either install significant overcapacity of RES or a complete back-up system of conventional units.

We note this understanding is incorrect. We note that our model employs a market equilibrium based approach rather than optimization approaches which is prevalent in power sector-only models. See Pietzcker et al. 2017 for a comparison of approaches to represent power sector dynamics across various process-based integrated assessment models (IAMs). See also Iyer et al. 2019 for an inter-model comparison exercise based on GCAM (the integrated assessment model used in our paper) and ReEDS (a power-sector-only model that is based on an optimization approach).

Specifically, in our model, even if there would be an unlimited zero cost solar PV option, other more expensive technologies would get share. In GCAM, electricity demand is shared out across different electricity technologies (conventional coal w/ and w/o CCS, IGCC w/ and w/o CCS, gas combined cycle w/ and w/o CCS, gas CT, solar PV, CSP, onshore wind, geothermal, and nuclear) according to a logit-based equation as follows:

$$s_{T,t} = \frac{\alpha_{T,t} p_{T,t}^{\gamma}}{\sum_{T=1}^N \alpha_{T,t} p_{T,t}^{\gamma}} \quad (\text{Equation 1})$$

Where $p_{T,t}$ is the levelized cost of the technology T in time period t (calculated as the sum of levelized capital and operating costs including fuel costs) and γ is an exogenous input shape

parameter. $\alpha_{T,t}$ are calibration parameters called “share-weights”. This formulation has an important property in that it assigns some market share to expensive technologies, which allows the model to avoid an unrealistic “winner take all” responses based on the notion that choices are based on other factors besides observed prices or that single observed prices do not represent the full variation in prices across applications. The leveled costs are in turn calculated using exogenously assumed capacity factors for every technology as follows.

$$p_{T,t} = \frac{C_{fuel}}{\eta} + \frac{1000 C_{capital}}{8760 CF} \times FCR + \frac{C_{O\&M, fixed}}{8760 CF} + C_{O\&M, variable} \quad (\text{Equation 2})$$

Where C_{fuel} is the fuel cost (\$/MWh); η is the power plant efficiency; $C_{capital}$ is the overnight capital cost (\$/kW), CF is the capacity factor of the technology in the investment segment, FCR is the fixed charge rate; $C_{O\&M, fixed}$ is the annual fixed O&M cost (\$/MW per year); $C_{O\&M, variable}$ is the variable O&M cost (\$/MWh) and 8760 is the number of hours in a year. The list of electric power generation technologies represented in the model and their input assumptions are documented in Muratori et al. 2017

In the first case, the presented changes in overall output due to climate change may not have a feedback on the overall capacity investments at all. In such a world they are way higher than the capacity formula in the methods section ($cap=gen/capfac$) would imply; simply to be able to account for weather variations, seasonality and other uncertainties. This would mean that the higher/lower average availability due to climate change would not necessarily translate into alterations on the capacity side at all.

We clarify that our paper does not account for weather fluctuations or seasonality. Our focus is on the long-term (century-scale). That said, the reviewer is correct that if seasonal and diurnal variation of demand were to be considered, capacity requirements would be higher. There are ongoing efforts both within our group and outside to improve sub-annual details in power sector representation in IAMs (e.g. Wise et al. 2019, Pietzcker et al. 2017).

However, we clarify that higher/lower average availability due to climate change would indeed translate into alterations on the capacity side. This is because, the higher/lower average availability of a renewable resource would translate into shifting of supply curves which would affect C_{fuel} in Equation 2 above. In our revised submission, we clarify this in the “Methods” section,, more specifically at the end of the “Climate impacts on renewables — model representation” subsection.

In the second case, there is a stable back-up system that would need to be built up regardless of the actual average RES injection as such a system would need to be designed for the no-RES-infeed hours anyway (as long as they don’t change due to climate change there is no feedback even if a higher total output of RES in a year would be possible). Thus whether altered RES injection would lead to altered RES capacity investments is not so clear.

This is a great point. As noted earlier, we do not model capacity implications of alternative RES injection as traditional power-sector-models do. That is part of ongoing research efforts across the IAM community. However, we note that our model does include a representation of

renewable intermittency. Like most IAMs, this is translated into costs that vary with share of renewables in the grid. As explained earlier, RES injection changes induced by climate change affect changes in supply curves which in turn affect the amount of renewables in the grid and hence the intermittency costs.

The calculation of intermittency costs is as follows. For each of the solar and wind technologies, GCAM includes, both a purely intermittent option and an option paired with dedicated storage. The representation of intermittency costs for intermittent renewables attempts to reflect the diminishing contributions to electric capacity reserves as the share of intermittent technologies in the grid increases. This is done with a simple exponential functional form that adds to the cost of adding new intermittent generation to secure additional capacity. The marginal reserve capacity requirement is computed as an exponential function of the share of intermittent renewable capacity in total electric power capacity as follows:

$$CapacityRatio = \frac{F_{max}}{1 + \exp\left(C * \frac{(x_{mid} - elecShare)}{\tau}\right)}$$

where *CapacityRatio* is the ratio of the reserve capacity to the capacity of intermittent renewable, F_{max} is the maximum capacity ratio (defaults to 1), C and τ are shape parameters (ratio of C and τ determine the steepness to reach F_{max}), *elecShare* is the share of total intermittent renewable capacity in total power capacity, x_{mid} is the value of *elecshare* at which $CapacityRatio = 0.5 * F_{max}$.

A representative curve with $F_{max}=1$; $C=5$ and $\tau=0.1$ and $x_{mid}=0.35$ is shown in Figure 1 below.

Figure 1. Reserve capacity ratio as a function of share of intermittent renewable capacity in total electric power capacity

At low shares of intermittent technologies, each additional intermittent unit requires a small fraction of reserve capacity. At a total share of 35%, the marginal intermittent unit counts 50%

toward capacity reserves. The cost of reserve capacity requirements is then calculated by assuming that the reserve capacity is provided by natural gas combustion turbines operating at a capacity factor of 5%.

I understand that this is nothing that the underlying model can easily include and is likely also not the objective of the paper. The point of altered RES potential and feedback to the overall energy system is valid nonetheless. But it would still require a significant rewrite of the discussion to properly account for this problem. And maybe the capacity part should be skipped completely and the focus should solely be on TWh numbers. It simply is not really possible to derive capacity projections for electricity systems with high shares of intermittent RES without going into a finer time resolution. And capacity projections based on yearly averages are way off by design.

The reviewer is correct in that incorporating seasonal and diurnal changes in demand and renewable intermittency is not the objective of the paper. In response to this comment, we have rewritten the discussion and methodology portions of the paper to better explain our approach and also sufficiently highlight some of the caveats (which in turn suggest a ripe area for future work). We however believe that our capacity results provide a useful initial insight into the potential order of magnitude of changes expected. In our revised submission, we highlight that our capacity results are intended to be only back-of-the-envelope calculations and model detailed modeling that better accounts for seasonal and diurnal variation of electricity demand and renewable supply could be a ripe area for future work. New text included in the “Revision” section reads:

“An important caveat of this analysis is that the version of GCAM used in this study represents electricity supply and demand on an annual mean basis assuming, for example, fixed exogenously-defined capacity factors for each power generation technology. Thus, the variability of electricity demand and load at seasonal and daily temporal scales is not considered, which has important implications for decisions on generation infrastructure. The challenge of continuously balancing supply and demand at such finer temporal scales becomes even more complex as the deployment of intermittent solar- and wind-based generation with limited dispatchability increases. Consequently, our analysis likely underestimates rates of capacity additions through 2100 because the annual average supply and demand electricity representation of GCAM smooths out short-term events of peak demand that require the highest electricity outputs. In light of this, our estimates of generation capacity and capital investments should be interpreted as a first-order approximation of the magnitudes of future needs that can be refined by follow-on studies. In this regard, there are ongoing efforts involving GCAM and other IAM groups to improve sub-annual details in power sector representation in IAMs (e.g., Wise et al.⁶⁴, Pietzcker et al.⁶⁵). Another consequence of its annual average electricity representation alongside simplifications of important processes is that GCAM cannot represent climate impacts at short timescales (e.g., seasonal scales). These characteristics also impose challenges for the representation of changes in climate variability and short-term extreme events within IAM frameworks. Hence, our study focuses on implications due to long-term (multi-decadal) mean climatological changes. Future investigation is needed to enhance GCAM modeling capabilities towards finer temporal scales and more detailed representations of power system dynamics.”

References:

- Pietzcker, R. C. *et al.* System integration of wind and solar power in integrated assessment models: A cross-model evaluation of new approaches. *Energy Economics* **64**, 583-599, doi:<https://doi.org/10.1016/j.eneco.2016.11.018> (2017).
- Iyer, G. C. *et al.* Improving consistency among models of overlapping scope in multi-sector studies: The case of electricity capacity expansion scenarios. *Renewable and Sustainable Energy Reviews* **116**, 109416, doi:<https://doi.org/10.1016/j.rser.2019.109416> (2019).
- Muratori, M. *et al.* Cost of power or power of cost: A U.S. modeling perspective. *Renewable and Sustainable Energy Reviews* **77**, 861-874, doi:<https://doi.org/10.1016/j.rser.2017.04.055> (2017).
- Wise, M. *et al.* Representing power sector detail and flexibility in a multi-sector model. *Energy Strategy Reviews* **26**, 100411, doi:<https://doi.org/10.1016/j.esr.2019.100411> (2019).

REVIEWERS' COMMENTS

Reviewer #1 (Remarks to the Author):

Dear Authors,

I acknowledge and appreciate the significant effort to address all the comments and add additional results. I think you have done a very good job of clarifying many sections and providing additional justification, explanation and policy insights. I think this analysis is robust, well described and provides valuable insights for modelling and policy.

Best regards,

Jen Cronin

Reviewer #2 (Remarks to the Author):

I have looked through the response to the reviews' comments. The authors have addressed all the question very well. I recommend the publication of the article.